# A key residue of the extracellular gate provides quality control contributing to ABCG substrate specificity

Jian Xia [1,9], Alexandra Siffert [1,10], Odalys Torres [2], Francesca Iacobini [1], Joanna Banasiak [3], Konrad Pakuła [3], Jörg Ziegler[4], Sabine Rosahl [5], Noel Ferro[6], Michał Jasiński [3,7], Tamás Hegedűs [2,8] ✉ & Markus M. Geisler [1,8] ✉

For G-type ATP-binding cassette (ABC) transporters, a hydrophobic "di-leucine motif" as part of a hydrophobic extracellular gate has been described to separate a large substrate-binding cavity from a smaller upper cavity and proposed to act as a valve controlling drug extrusion. Here, we show that an L704F mutation in the hydrophobic extracellular gate of Arabidopsis ABCG36/PDR8/PEN3 uncouples the export of the auxin precursor indole-3-butyric acid (IBA) from that of the defense compound camalexin (CLX). Molecular dynamics simulations reveal increased free energy for CLX translocation in ABCG36$^{L704F}$ and reduced CLX contacts within the binding pocket proximal to the extracellular gate region. Mutation L704Y enables export of structurally related non-ABCG36 substrates, IAA, and indole, indicating allosteric communication between the extracellular gate and distant transport pathway regions. An evolutionary analysis identifies L704 as a *Brassicaceae* family-specific key residue of the extracellular gate that controls the identity of chemically similar substrates. In summary, our work supports the conclusion that L704 is a key residue of the extracellular gate that provides a final quality control contributing to ABCG substrate specificity, allowing for balance of growth-defense trade-offs.

ABC proteins represent an ancient and versatile transport system operating in all living organisms[1–3]. Most ABC transporters are primary pumps, which utilize the energy of ATP-dependent hydrolysis to transport various substrates across cellular membranes[4]. Ectopic or dysregulated over-expression of certain ABC transporters often contributes to multidrug resistance (MDR) phenomena due to non-selective translocation of various drugs and thus can be a severe complication for treating cancer as well as for microbial and parasitic infections[5]. Three human ABC transporters, ABCB1 (Multidrug resistance protein 1, MDR1; P-glycoprotein 1, Pgp1), ABCC1 (Multidrug resistance-associated protein 1, MRP1), and ABCG2 (Breast cancer resistance protein, BCRP), are thought to commonly promote or cause MDR in a variety of therapeutic cancer settings.

[1]University of Fribourg, Department of Biology, Fribourg, Switzerland. [2]Department of Biophysics and Radiation Biology, Semmelweis University, Budapest, Hungary. [3]Department of Plant Molecular Physiology, Institute of Bioorganic Chemistry, Polish Academy of Sciences, Poznań, Poland. [4]Department Molecular Signal Processing, Leibniz Institute of Plant Biochemistry, Halle (Saale), Germany. [5]Department Biochemistry of Plant Interactions, Leibniz Institute of Plant Biochemistry, Halle (Saale), Germany. [6]Ferro CBM, Friedrich-Vorwerk-Straße 13-15, Tostedt, Germany. [7]Department of Biochemistry and Biotechnology, Poznan University of Life Sciences, Poznań, Poland. [8]Biophysical Virology Research Group, HUN-REN-SU, Budapest, Hungary. [9]Present address: Institute of Agricultural and Nutritional Sciences, Martin Luther University Halle-Wittenberg, Halle (Saale), Germany. [10]Present address: Department of Plant and Microbial Biology, University of Zürich, Zürich, Switzerland. ✉e-mail: hegedus.tamas@hegelab.org; markus.geisler@unifr.ch

In higher plants, the ABC family is expanded twice to four times compared to other organisms, including mammals or their ancestral microalgae[6,7]. This has led to the concept that ABC transporters multiplied during evolution and acquired novel functions that allowed plants to adapt to a sessile and terrestrial lifestyle[8]. This idea is indirectly supported by the finding that many plant ABC transporters transport growth hormones, like auxins and cytokinins, playing a crucial role in post-embryonic development[7]. Up to date, research on plant ABCs has focused on ABCBs because a subset was shown to catalyze the transport of auxin and to be determinants of plant development[9–12]. Recently, the focus has shifted toward PLEIOTROPIC DRUG RESISTANCE (PDR)-type ABC transporters, which are full-size ABC transporters of the G-type and found mostly in plants and fungi[6,13]. One of the most investigated plant PDR to date is ABCG36/PENETRATION3 (PEN3)/PDR8, for which a range of structurally unrelated, putative substrates were identified[13,14]. Besides the auxin precursor, indole-3-butyric acid (IBA[13]), ABCG36 was shown to export two indolic compounds functioning in defense signaling[15]. Moreover, ABCG36 was characterized as an exporter of the antimicrobial, indolic compound camalexin (CLX[14]). Based on the physiological importance of these substrates (IBA and CLX) for Arabidopsis, it was suggested that ABCG36 functions at the cross-road between plant growth and pathogen defense, which represents one of the significant tradeoffs that plants have to balance[16]. The role of ABCG36 in these two programs was recently shown to be regulated via protein phosphorylation by the leucine-rich repeat receptor kinase QIAN SHOU KINASE1 (QSK1). ABCG36 phosphorylation by QSK1 unilaterally represses IBA export, promoting CLX export by ABCG36, conferring pathogen resistance[14]. ABCG36 might thus play an essential role during the regulation of plant defense-growth trade-offs. This is underlined by the finding that the physiological consequence of prolonged defense is growth inhibition[16,17] because essential resources are thought to be allocated toward defense[16]. On the other hand, such a regulatory role might also be of importance from a metabolic point of view because both indolic defense compounds and IAA-intermediates (such as IBA) are Trp-derived[18–20].

Significant resources were dedicated to uncovering the structure of ABCG proteins to understand their function and substrate recognition. The first high-resolution ABCG crystal structures of the human half-size transporter heterodimer, ABCG5/ABCG8[21], and cryo-EM structures of the ABCG2 homodimer[22,23] revealed that despite an overall similar transport mechanism, ABCG transporters embody a transmembrane domain structure later described as type V[24], which is significantly different compared to type IV transporters, represented by ABCB, ABCC, and ABCD subfamily members. In the core of the substrate promiscuous ABCG2 homodimer, a hydrophobic "di-leucine motif" was described to separate a larger lower intracellular cavity - serving as a binding region for substrates and inhibitors, from a smaller upper cavity[22]. Later, this hydrophobic "leucine plug" was suggested to act as a valve to control drug extrusion[25]. Furthermore, the ABCG2 structure contains a polar "extracellular roof" with a compact semi-closed architecture, which supposedly acts as a barrier for substrate release from the upper cavity[25]. In contrast, ABCG5/G8 exhibits a collapsed cavity of insufficient size to hold the substrates[26] and leucine residues of the putative valve are substituted by hydrophobic residues phenylalanine and methionine, respectively. The functional implications of these residue differences are unclear, but the larger aromatic phenylalanine side chain was speculated to be critical for sterol selectivity[26,27].

The structure of the yeast pleiotropic drug resistance transporter PDR5 was resolved by single-particle cryo-EM[28]. In contrast to other ABCGs, one-half of the transporter remains nearly invariant throughout the cycle, and the other half undergoes changes transmitted across inter-domain interfaces to support the transport process by peristaltic motions. In an AlphaFold2-derived structure of PDR-type ABCG46

from the model crop plant *Medicago truncatula*, an unusually narrow transient access path to the central cavity constitutes an initial filter responsible for the selective translocation of phenylpropanoids[29]. Mutation of F562 located in TH2 (Transmembrane Helix 2) affected the viability of the transient access path through helix rearrangements, profoundly affecting the substrate selectivity of Medicago ABCG46 toward the phenylpropanoids, 4-coumarate, and liquiritigenin[29]. Recently, three independent studies published cryo-EM structures of the Arabidopsis half-size transporter, ABCG25, working as a homodimeric exporter of the plant hormone abscisic acid (ABA)[30–32]. The interface between ABA and ABCG25 is mainly formed by hydrophobic interactions with the residues of TH2 and TH5a[31]. Interestingly, here ABA translocation seems to be controlled by two gates: a "cytoplasmic gate" formed by residue F453 from TH2/TH2', and an "apoplastic gate" formed by Y565 from TH5a/TH5a'[31]. The latter is a functional homolog of the hydrophobic "leucin plug" described for human ABCG2[1,22,25].

Based on pathogen susceptibility and IBA sensitivity assays, a specific point mutation in *ABCG36/PEN3* of Arabidopsis was described to uncouple ABCG36 functions in IBA-stimulated root growth, callose deposition, and pathogen-inducible salicylic acid accumulation from ABCG36 activity in pathogen defense. It was suggested that this uncoupling in the *abcg36-5/pen3-5* allele might be caused by a lack of export of major relevant defense compounds (such as CLX), while IBA export was thought to be preserved[33]. Interestingly, the *abcg36-5* allele encodes for a L704F change in elongation of TH5 (misannotated as TH4 in Lu et al. (2015)) of ABCG36.

In this study, we uncover that uncoupling of ABCG36-mediated function in IBA-stimulated root growth from its transport role in defense in the *abcg36-5* allele (L704F) is due to unilateral loss of CLX transport capacity, while IBA transport is preserved. L704 (together with F1375) is a *Brassicaceae* family-specific key residue of an extracellular, hydrophobic gate that provides quality control of substrate specificity in ABCG36.

## Results

### L704F mutation in ABCG36 uncouples export of IBA and camalexin

Recent work has established the engagement of multiple ABCG36 substrates in different ABCG36-dependent critical biological processes, including growth and defense[13–15,33–39]. The *abcg36-5* allele uncouples ABCG36 functions in IBA-stimulated root growth from ABCG36 activity in extracellular defense[33], but the underlying molecular determinants are unclear. This unique IBA-uncoupling of the *abcg36-5* allele was originally found by using an IBA-mediated primary root growth inhibition phenotype in the presence of toxic μM concentrations[14,19,33].

In order to verify these findings, we replicated the originally described detoxification assay[19] under our conditions, and included the *abcg36-4* and *abcg36-6* loss-of-function alleles as negative controls. The first is a T-DNA insertional null allele, while the second carries an A1357V substitution in TH11[33] (Supplementary Fig. 3e). As shown before[33], *abcg36-4* and *abcg36-6* alleles but not *abcg36-5* exhibited reduced root lengths on IBA in comparison with their Wt backgrounds (Supplementary Fig. 1a, b). A repetition of this assay on cytotoxic concentrations of CLX[14], indicated as expected, a hypersensitivity for *abcg36-4*, *abcg36-6*, and *abcg36-5* alleles in comparison to their corresponding Wt backgrounds, Col-0 and Gl1, respectively (Supplementary Fig. 1c, d).

These results suggested that selective uncoupling of IBA detoxification was most likely due to the lack of CLX export capacities on *abcg36-5*[13]. In order to test this possibility, we first quantified IBA and CLX export from leaf mesophyll protoplasts prepared from the leaves of Wt and mutant plants after loading of ³H-IBA or ³H-CLX, respectively[12,14]. Results indicated significantly reduced CLX but not IBA export from *abcg36-5* cells, while both *abcg36-4* and *abcg36-6*

showed significantly reduced IBA and CLX export (Fig. 1a, b). For the chemically related auxin, IAA, previously reported not to be transported by ABCG36[14,40], as well as the diffusion control benzoic acid (BA) and indole, no significant differences to Wt were found (Fig. 1c; Supplementary Fig. 2a, b).

To secondly verify these Arabidopsis data in a heterologous system, allowing to exclude off-target effects by mutational events, we functionally expressed Wt and mutant versions, which contained the exact point mutations in *ABCG36-GFP* as in the described *abcg36* alleles, in tobacco by using agrobacterium-mediated leaf transfection[41]. As expected, the L704F (*abcg36-5*) variant had no effect on ABCG36-mediated IBA but abolished ABCG36-mediated CLX export to vector control level (Fig. 1d, e). In agreement, IBA loading into ABCG37[L704F]-expressing protoplasts was not significantly different from ABCG protoplasts, while ABCG37[L704F] loading was identical to Wt ABCG36 (Supplementary Fig. 2c, d). Further, the A1357V (*abcg36-6*) allele was not able to transport either of the substrates (Fig. 1d, e). Finally, combining both amino acid substitutions had no significant additional impact, excluding additive effects (Fig. 1d, e). As before in Arabidopsis, mutations had no significant effect on the export of IAA, BA, and indole as being non-ABCG36 substrates (Fig. 1f, Supplementary Fig. 2e, f). Also, ABCB36 mutation had no significant impact on PM expression levels in tobacco based on quantification of confocal images (Fig. 1g, h) and Western blot analyses (Supplementary Fig. 2j)

We also tested the effect of these mutations on ABCG36 ATPase activities measured on isolated microsomes prepared from transfected tobacco at pH 9, where ATPase activity of H+-ATPases is negligible, excluding indirect effects[14]. As shown before,[14] ABCG36 ATPase activity was significantly higher than the vector control and found to be stimulated by its substrates (IBA and CLX) but not by non-transported indolic compounds (IAA or indole; Fig. 1i). As expected, mutation of the N-terminal Walker A motif (K210L) diminished ABCG36 ATPase activity to vector control level (Fig. 1i). ATPase activity of ABCG36[L704F] was comparable to Wt and showed likewise a stimulation by IBA and CLX. Activities of ABCG36[A1357V] were slightly enhanced but lacked substrate stimulation. The latter is of interest because recently L554A substitution of the homologous residue in human ABCG2 resulted in an unstable protein, while L555A exchange showed lower substrate-stimulated ATPase activity, but twice the translocation activity of the wild-type transporter[22].

Thirdly, in order to exclude ecotype-specific (Col-0 vs. Gl1 Wt) differences in root elongation and/or transport, we complemented the T-DNA insertion loss-of-function allele *abcg36-4* with single and double point-mutated, genomic versions of *ABCG36* (*ABCG36:ABCG36-GFP*) expressed under its native promoter[39]. Like Wt[14], two independent lines of *ABCG36[L704F]* fully complemented the hypersensitivity of *abcg36-4* roots toward IBA and its incompetence of IBA export (Fig. 2a, c), while this was not found for CLX (Fig. 2b, d). As seen with chemically generated point mutations (Fig. 2a, d;[33]), *ABCG36[A1357V]* and *ABCG36[L704F, A1357V]* lines did not complement the *abcg36-4* mutant. As before, these mutations did not significantly alter PM expression and polarity of ABCG36 in Arabidopsis based on quantification of ABCG36-GFP PM signals in Arabidopsis roots (Fig. 2e, f).

These datasets clearly demonstrate that uncoupling of ABCG36-mediated function in IBA-stimulated root growth from its transport role in defense in the *abcg36-5* allele (L704F) is due to unilateral loss of CLX transport capacity, while IBA transport is preserved.

## ABCG36 might own a signaling function during infection
For *abcg36-5* and *abcg36-6* alleles, mixed sensitivities toward shoot pathogens have been reported: while both alleles retain susceptibility to the host-adapted fungus, *Golovinomyces orontii*, they are fully defective in defense to non-adapted powdery mildews[33]. In order to test how these point mutations perform upon root infection, we

treated them with *Fusarium oxysporum*, a well-characterized, root vascular fungal pathogen that causes wilt disease in several plant species, including *Arabidopsis thaliana*[42]. This pathosystem was recently used to uncover a phospho-switch balancing IBA and CLX transport capacities[14]. In line with reduced CLX export capacities caused by L704F and A1357V mutations (Figs. 1, 2), both alleles were like KO mutants[14] hyper-susceptible toward *Fusarium* infection based on quantification of leaf symptoms (Fig. 3a, b).

In order to be able to quantify exported ABCG36 substrates upon infection in these alleles, we changed back to a leaf pathosystem using established drop-inoculation of leaves with the oomycete *Phytophthora infestans*[14,15]. As described previously, water controls of both Wt ecotypes showed very low surface levels of CLX[14,15] that were not significantly different from *abcg36* alleles (Fig. 3c). However, surface drops that contained *P. infestans* zoospores revealed highly enhanced CLX levels in Wt compared to the non-infected control, while CLX concentrations collected from *abcg36-5* and *abcg36-6* leaves were strongly reduced in comparison to the infected Wts, respectively (Fig. 3c).

Because it is very difficult to reliably detect IBA in the infection droplets, most likely due to its low abundance[18], we looked at the ABCG36 substrate, 4-methoxyindol-3-yl-methanol (4MeOI3M), which, based on competition experiments is transported in an IBA-specific manner[14,15]. In agreement with IBA transport experiments (Figs. 1, 2), 4MeOI3M export upon infection was significantly reduced in the transport-dead *abcg36-6* allele but not in *abcg36-5* (Fig. 3d) shown to retain IBA export (Figs. 1, 2) and IBA-mediated root elongation[33].

Reduced CLX and 4MeOI3M export for *abcg36-6* but only a reduction of CLX export for *abcg36-5* in comparison to the corresponding Wt seems to be intuitively correct but is on a first view in conflict with previously published data showing strongly enhanced CLX and reduced 4MeOI3M presence, respectively, in infection droplets for both *abcg36-3* and *abcg36-4* T-DNA insertional, loss-of-function mutants upon infection[14]. A rationale for this behavior was provided by the finding that ABCG40, a putative CLX and a bona fide IBA exporter[34], was strongly upregulated on the protein level in the *abcg36-3* allele upon infection[14]. A reasonable and attractive explanation for this contradiction can be found in the concept that this over-compensation by ABCG40 (and eventually other related isoforms[14]) might not take place in mutated, PM expressed versions of ABCG36 (as found for *abcg36-5* and *abcg36-6* alleles; Fig. 3c) but only in the absence of the ABCG36 protein as proven for *abcg36-3* and *abcg36-4* null alleles[33].

In summary, this dataset supports on one hand the unilateral absence of CLX transport for the *abcg36-5* allele in a root-based pathosystem but on the other indicates a previously unknown signaling function for ABCG36 in an interplay with closely related ABCG isoforms.

## The L704 mutation is part of the extracellular gate and influences the transport process primarily by allosteric effects
To provide a mechanistic explanation for the ability of ABCG36 to discriminate IBA and IAA and for the unilateral substrate discrimination between IBA and CLX found for the L704F variant, we employed various in silico methods. Since in silico docking of substrates to ABCG2 suggested several drug-binding spots along the substrate translocation pathway[43], we performed metadynamics simulations to study the translocation process. However, because such a large protein system likely prohibits reaching full convergence, we supplemented these calculations with pulling simulations.

The inside-closed AlphaFold2 structure of ABCG36 was used in all of our simulations since this structure is the closest to the transport-competent conformation (Supplementary Fig. 3[44]). Interestingly, sequence analysis and mapping of the *abcg36-5* mutation on this ABCG36 structure confirmed that L704 is part of the "leucine valve"

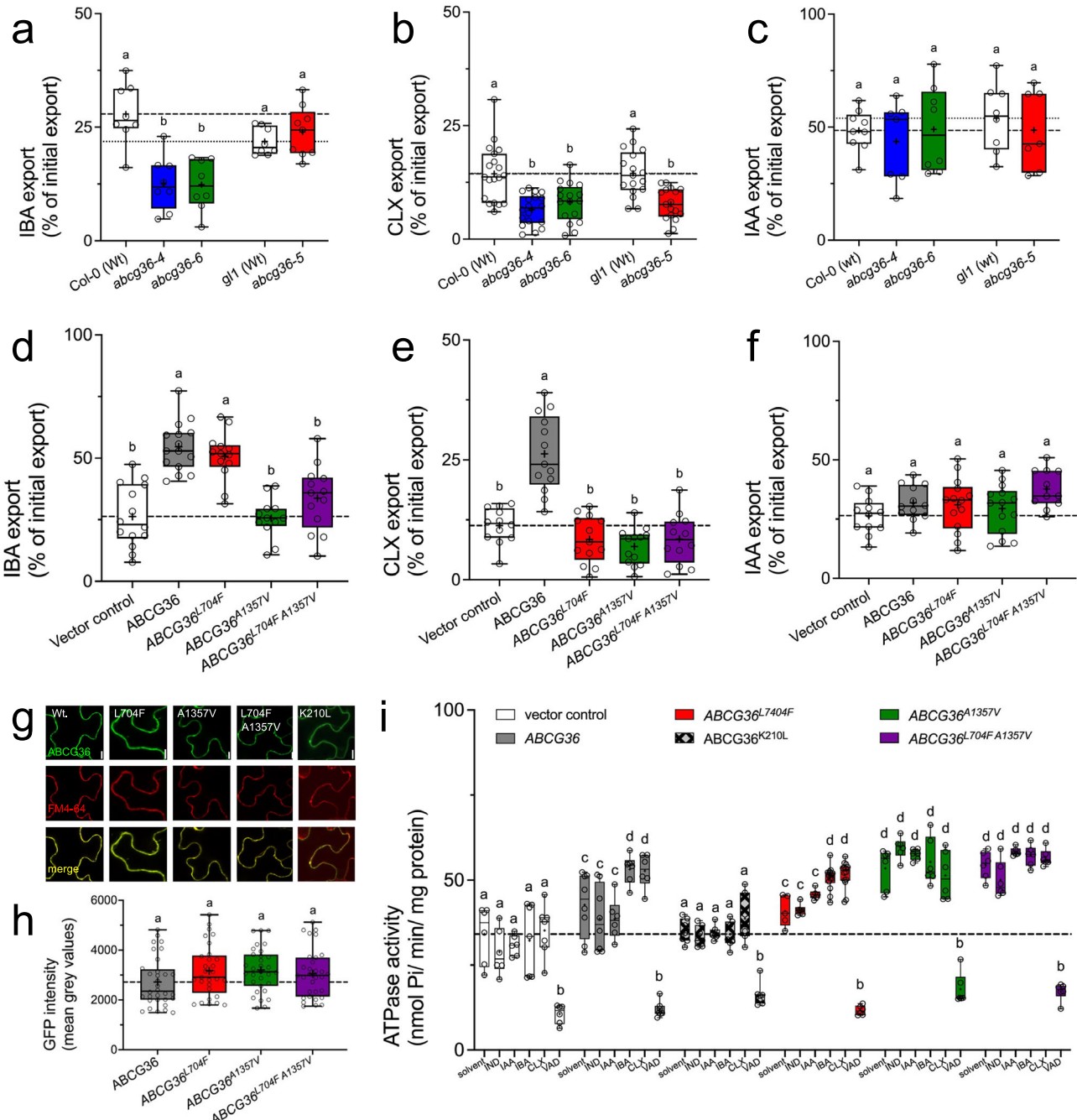

**Fig. 1 | L704F mutation in ABCG36 uncouples IBA from camalexin export.** IBA (**a**) CLX (**b**) and IAA (**c**) export from *Arabidopsis* protoplasts prepared from indicated *ABCG36* loss-of-function alleles. Significant differences (*p* < 0.05) of means ± SE (*n* = 8 independent protoplast preparations) were determined using Ordinary One-way ANOVA (Tukey's multiple comparison test) and are indicated by different lowercase letters. IBA (**d**) CLX (**e**) and IAA (**f**) export from *N. benthamiana* protoplasts after transfection with indicated mutant versions *ABCG36*. Significant differences (*p* < 0.05) of means ± SE (*n* = 14 independent protoplast preparations) were determined using Ordinary One-way ANOVA (Tukey's multiple comparison test) and are indicated by different lowercase letters. Confocal imaging (**g**) and quantification (**h**) of GFP-tagged versions of mutated *ABCG36* after transfection of *N. benthamiana* leaves. Short treatment of FM4-64 was used as PM markers; bar, 50

μm. Significant differences (*p* < 0.05) of means ± SE (*n* = 25 epidermal cells) were determined using Ordinary One-way ANOVA (Tukey's multiple comparison test) and are indicated by different lowercase letters. **i** ATPase activity of microsomal fractions prepared from tobacco leaves transfected with vector control or indicated mutant versions of *ABCG36* measured at pH 9.0 in the presence of 50 μM indole, IAA, IBA, CLX, or ortho-vanadate. Significant differences (*p* < 0.05) of means ± SE (*n* = 3 independent transfections and microsomal preparations) were determined using Ordinary Two-way ANOVA (Tukey's multiple comparison test) and are indicated by different lowercase letters. Data are presented as box-and-whisker plots, where median and 25th and 75th percentiles are represented by the box itself and the middle line, respectively; means are indicated by a "+". Source data is provided as a Source Data file.

terminating the substrate funnel, which is formed by extensions of transmembrane helices TH2, TH5, TH8, and TH11 (Figs. 4 and 5a; Supplementary Fig. 7[25]). F703/F1374 and L704/F1375 of ABCG36 correspond to L554 and L555 of human ABCG2, respectively

(Supplementary Fig. 7a) and protrude from TH5 and bTH11, respectively (Supplementary Fig. 7). A1357, the cause of the *abcg36-6* mutation, is also part of TH11 but is located towards the substrate entry side (Supplementary Fig. 3e).

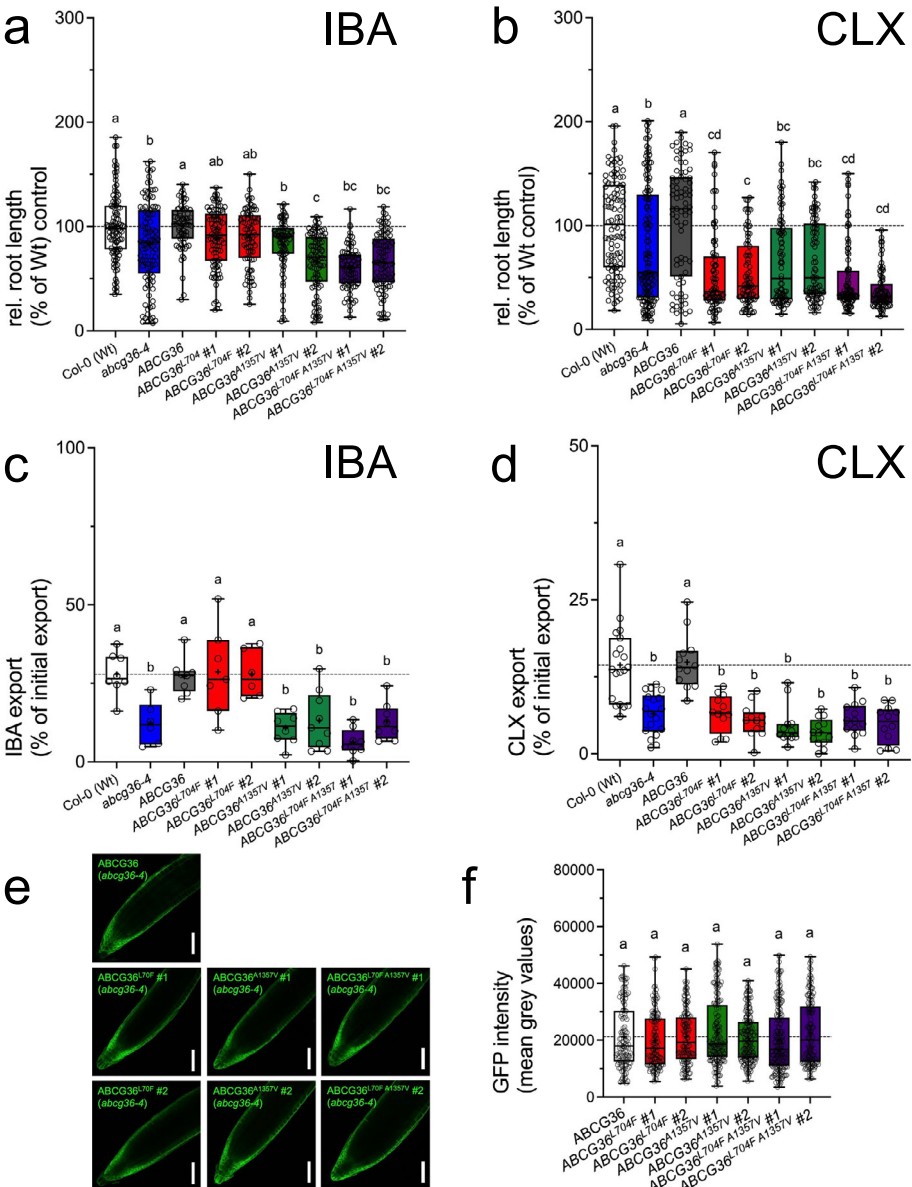

**Fig. 2 | Uncoupling of IBA from camalexin export is ecotype-independent.** Relative root length of two independent Arabidopsis *abcg36-4* lines complemented with indicated mutant versions of *ABCG36* grown for 12 days on 7.5 µM IBA (**a**) or 5 µg/ml CLX (**b**); Wt growth is set to 100%. Significant differences ($p < 0.05$) of means ± SE ($n = 3$ independent experiments with each 10 seedlings) were determined using Ordinary One-way ANOVA (Tukey's multiple comparison test) and are indicated by different lowercase letters. **c**, **d** IBA (**c**) and CLX (**b**) export from two independent Arabidopsis *abcg36-4* lines complemented with indicated mutant versions of *ABCG36*. Significant differences ($p < 0.05$) of means ± SE ($n = 8$ independent protoplast preparations) were determined using Ordinary One-way ANOVA (Tukey's multiple comparison test) and are indicated by different lowercase letters. Confocal imaging (**e**) and quantification (**f**) of Arabidopsis *abcg36-4* lines complemented with indicated mutant versions of *ABCG36*; bar, 50 µm. Significant differences ($p < 0.05$) of means ± SE ($n = 100$ root cells) were determined using Ordinary One-way ANOVA (Tukey's multiple comparison test) and are indicated by different lowercase letters. Data are presented as box-and-whisker plots, where median and 25th and 75th percentiles are represented by the box itself and the middle line, respectively; means are indicated by a "+". Source data is provided as a Source Data file.

Ab initio calculations using density functional theory (DFT) allowed to calculate electronic structural deformation and the electric field lines of IBA, CLX and IAA before and after binding to the ABCG36 substrate pocket (Fig. 4c, d). For IBA and CLX, the electronic structure remained stable though revealing changes in field lines. In contrast, the binding of IAA to the ABCG36 pocket leads to significant changes in the electronic structure of IAA, particularly in its side chain (Fig. 4c, d). This is accompanied by a partitioning of the electronic deformation within the interatomic bonding structure, resulting in a decrease in internal molecular stability. The buffer region, which corresponds to the coupling between the side chain and the IAA ring system, also exhibits the

same phenomenon. The electronic structure of the internal bonds is not retained after binding; however, this region is crucial for the biological activity of auxins[45]. Essentially, the molecule is no longer able to fit properly into the binding pocket, or the pocket does not allow the IAA to exist as a complete molecule within it, or both. The electronic binding energies between ABCG36 residues of the central pocket and the ligands IBA, CLX, and IAA are −19.23, −8.11, and 34.08 kcal/mol, respectively. The positive energy for IAA essentially rules out IAA as an ABCG36 ligand that can be transported by ABCG36.

Next, we quantified specific binding of radiolabeled IBA and CLX employing an established microsome-based assay[46]. Our results using

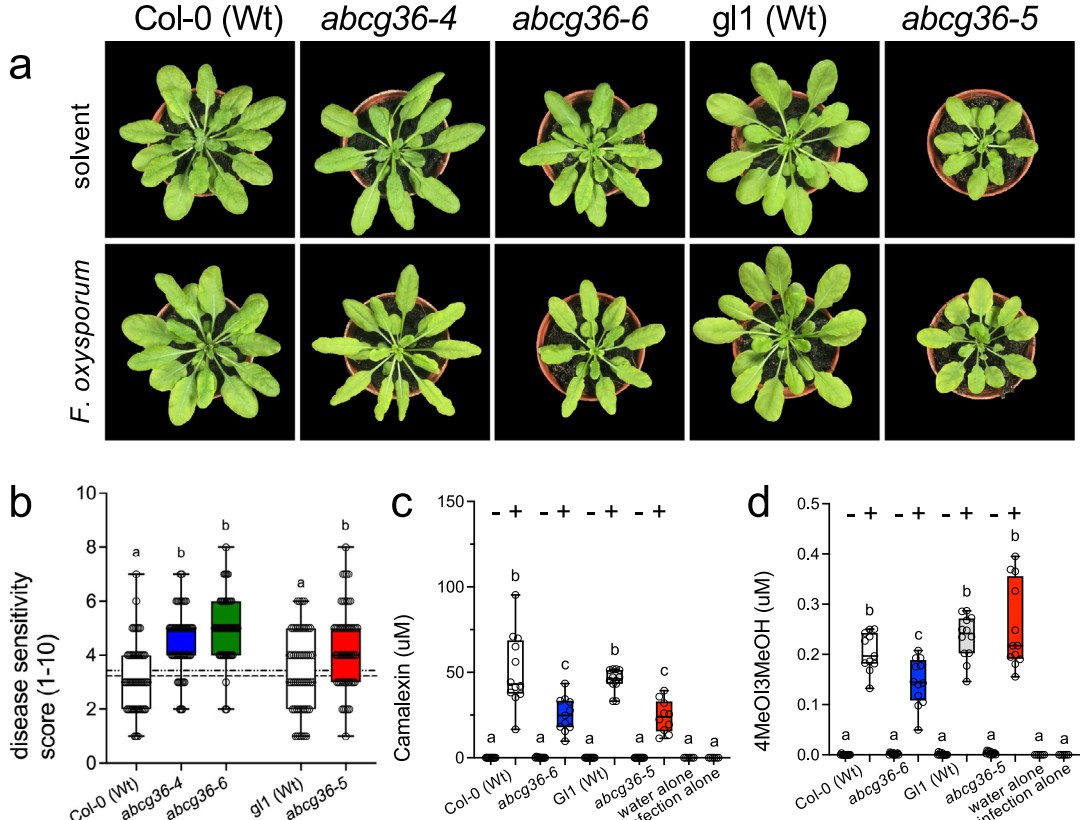

**Fig. 3 | ABCG36 has a sensor-like function during infection.** 5-week-old plants grown on soil were watered with buffer (untreated control) or with Fo (Fo699; $10^7$ conidia/ml). Representative plants are pictured (**a**) and disease symptoms were quantified (**b**) using a scale from 0-10 [14]. Significant differences ($p < 0.05$) of means ± SE ($n = 3$ independent infection series with each 15 single plants) were determined using Ordinary One-way ANOVA (Tukey's multiple comparison test) and are indicated by different lowercase letters. **c** Extracellular levels of camalexin

(**c**) and 4MeOI3M (**d**) on *P. infestans*-inoculated *Arabidopsis* leaves determined by non-targeted UPLC-ESI-QTOF-MS. Significant differences ($p < 0.05$) of means ± SE ($n = 7$ with each 30 droplets) were determined using Ordinary One-way ANOVA (Tukey's multiple comparison test) and are indicated by different lowercase letters. Data are presented as box-and-whisker plots, where median and 25th and 75th percentiles are represented by the box itself and the middle line, respectively; means are indicated by a "+".

microsomes prepared from Arabidopsis Wt and mutants clearly support IBA and CLX binding to ABCG36 based on significant reductions for the *abcg36-4* loss-of-function allele (Fig. 4e, f). In agreement with our transport data, *abcg36-6* and *abcg36-5* microsomes revealed reduced binding of both IBA and of CLX, respectively, but differences were non-significant. Therefore, mutations in the extracellular gate do either only mildly affect drug binding or observed reductions in substrate binding are masked by other membrane proteins, including described ABCG-type IBA exporters shown to transport IBA and CLX[14].

The 1D free energy surface (FES) of IBA translocation calculated from metadynamics simulations along the selected reaction coordinate (collective variable, Supplementary Fig. 4) exhibited an overall downhill process with some barriers from the intra- to extracellular direction for both Wt and L704F mutant versions of ABCG36 (Fig. 4g). In contrast, the FES from Wt ABCG36 CLX simulations exhibited low values at the intracellular ends of helices (around dz of -1 nm), suggesting that this molecule can engage with this protein conformation easier than IBA (Fig. 4h). The FES for CLX from the L704F ABCG36 simulation revealed a slightly elevated energy barrier between −1 and −0.5 nm when compared to Wt, suggesting a more energy intensive access of CLX to the central pocket (Fig. 4h), a potential explanation of abrogated transport. Metadynamics simulations with a set of mutations and the four molecules (Fig. 5g and Supplementary Fig. 5) indicated that differences in interactions with residues in the inner membrane leaflet and in the central binding pocket may play a primary role in influencing substrate recognition.

Since the FES around the extracellular gate (>1.8 nm) was not entirely conclusive about the role of this region in the transport, we performed pulling simulations to identify critical residues along the transport pathway interacting with the studied small molecules (Supplementary Fig. 4). To better visualize the results of our pulling simulations, contact frequencies were projected on ABCG36 structures using color encoding. The overall interaction pattern of transported substrates, such as IBA for Wt or L704F and CLX for Wt were similar (Fig. 4i, j). In contrast, the non-substrate interactions of CLX for L704F were characteristic for residues in the inner leaflet rather than in the central binding pocket (Fig. 4j) and are in good agreement with metadynamics data suggesting a higher energy state in the central binding pocket for non-substrates (Supplementary Fig. 6). The slightly higher contact frequencies (higher forces; e.g. for residues I585 and F698) observed in the case of IBA pulling from the central binding pocket compared to CLX pulling in Wt ABCG36 (Fig. 4i, j) also agree with the DFT binding energy calculations (Fig. 4c, d).

Importantly, IAA, which is structurally very similar to IBA (Fig. 4a), also shows lower interaction frequencies within the binding pocket proximal to the extracellular gate region when it is not a substrate (Wt ABCG36, Fig. 1[13–15]) and higher interactions frequencies when acting as a substrate (L704Y ABCG36, Supplementary Fig. 6). Moreover, the FES from the metadynamics simulation with this setup is also decreased in the 0 to 0.8 nm region, compared to Wt (Supplementary Fig. 5).

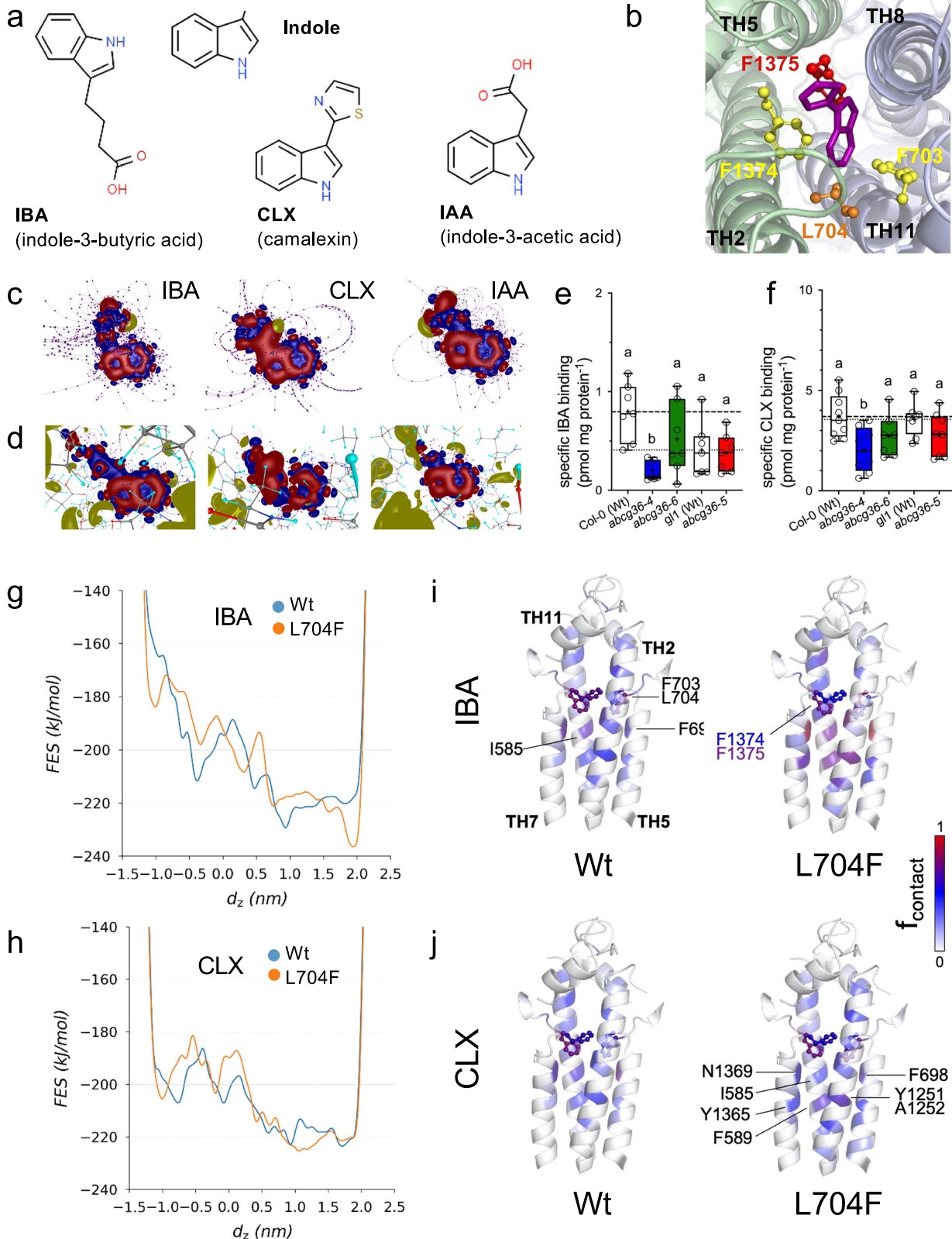

In conclusion, our simulations highlight that substrate transloca-tion is likely controlled by different regions, including both the entrance region (Fig. 4g, h) and the central binding pocket (Fig. 4i, j). Interestingly, mutations in the extracellularly located gate can affect the energetics of the substrate interaction with distant protein regions (e.g., beginning of the translocation pathway, 1–2 nm from the extra-cellular gate, Fig. 4g), suggesting allosteric coupling.

## L704Y mutation in ABCG36 leads to a broadening of substrate specificity toward indolic compounds

In order to further challenge this apparent quality control function of the extracellular gate, we further investigated the transport specificity of L704 mutated versions of ABCG36 by site-directed mutagenesis. For choosing relevant mutations, we employed an amino acid property approach[47]. Interestingly, like L704F, all tested mutations of L704

**Fig. 4 | L704 is part of the extracellular gate of ABCG36 and L704F increases free energy surface for camalexin in the entrance region. a** Chemical formulas for potential ABCG36 substrates employed in this study (downloaded from PubChem (https://pubchem.ncbi.nlm.nih.gov)). **b** THs of ABCG36 are shown from the extracellular space. Extracellular gate residues are highlighted by sticks and balls. Docked IBA is shown using stick representation. Electronic structural deformation, the electric field lines of IBA, CLX, and IAA before (**c**) and after binding to the ABCG36 substrate pocket (**d**). The electronic binding energies and the results displayed here were obtained by ab initio calculations using DFT methods. **e, f** Specific ³H-IBA and ³H-CLX binding to indicated Arabidopsis microsomes calculated as the difference between total and unspecific substrate binding determined in the absence (total) and presence of a 1000-fold excess of non-radiolabeled substrate concentrations (unspecific), respectively. Significant differences ($p < 0.05$) of means ± SE ($n = 3$ independent microsomal preparations with each 3 technical replica) to the corresponding Wt were determined using Ordinary One-way ANOVA (Tukey's multiple comparison test) and are indicated by different lowercase letters. Data are presented as box-and-whisker plots, where median and 25th and 75th percentiles are represented by the box itself and the middle line, respectively; means are indicated by a "+". Source data are provided as a Source Data file. Free energy surfaces (FES) were calculated from metadynamics simulations with complexes of IBA (**g**) and CLX (**h**) with Wt and L704F ABCG36, respectively. Pulling of IBA (**i**) and CLX (**j**) from the central binding pocket to the extracellular space were performed in biased simulations with Wt and L704F ABCG36, respectively. Frequencies of contacts between the small molecules and the protein are plotted.

retained IBA transport capacities, while this was not the case for CLX export (Fig. 5a, b). Exchange of leucine against alanine and positively and negatively charged arginine and aspartate, respectively, also abolished CLX export, while substitution for the aromatic phenylalanine or tyrosine did not (Fig. 5b).

We included in our transport analyses also IAA and indole because they are, first, analogous to IBA or contain the indole core of all tested substrates and, second, they were neither transported by Wt nor L704F versions of ABCG36 (Fig. 5c, d; Supplementary Fig. 2). Remarkably, we found a gain-of-IAA transport for polar but uncharged serine and the hydrophobic aromatic residue, tyrosine, but not for the closely related tryptophane (Fig. 5c). Interestingly, L704 exchange to tyrosine was the only mutation that enabled indole transport (Fig. 5d). Importantly, all mutations of L704 (except L704A) did not alter transport of the organic acid, BA (Supplementary Fig. 2g), commonly used as a diffusion control[12]. Further, all mutations resulted in stable proteins on the PM and expression levels were comparable (Fig. 5e, f) allowing us to conclude that altered transport capacities were the direct cause of these mutations.

Next, we aimed to exclude that L704Y exchange did result in an uncoupling of ATPase activity and transport, leading eventually to a permanently open conformation and thus an unspecific transporter. To do so, we performed competition experiments of IAA and indole transport using inside-out prepared from tobacco leaves infiltrated with ABCG36^L704Y. As shown before for Wt ABCG36, transport of IBA and CLX[14], IAA, and indole transport was completely abolished by 100-fold access of CLX, IBA, or indole/IAA, respectively (Supplementary Fig. 2h, i), indicating that ABCG36^L704Y is a functional ABC transporter.

The FES from metadynamics simulations with L704Y ABCG36 in the presence of indole revealed a reduced barrier in the -0.8 to 0.2 nm range compared to the Wt version (Fig. 5g). Interestingly, indole contacts with WT and L704Y ABCG36 calculated from pulling simulations (Fig. 5h) displayed trends strikingly similar to those observed for CLX contacts with L704F and WT, respectively (Fig. 4). Indole, a non-substrate for Wt ABCG36, preferentially interacted with residues located in a more intracellular direction (Fig. 5h) rather than with the L704Y residues positioned at the center of the binding pocket (toward the extracellular direction; Fig. 5i). This observation aligns well with our finding that substrates predominantly interact with binding pocket residues near the extracellular gate (Fig. 4 and Supplementary Figs. 4, 6).

In summary, this mutational work further supports the conclusion that L704 is a key residue of the extracellular gate that provides quality control, contributing to substrate specificity of ABCG36 toward a few indolic compounds. Exchange of the small, hydrophobic leucine to the hydrophobic but aromatic tyrosine broadens the substrate specificity slightly and allows for the export of structurally related, IAA and indole.

## L704 is a family-specific key residue of the extracellular gate

he *Brassicaceae* (indolic compounds) but use an identical set of growth hormones (here: IBA), which is also an indole[18,48]. Of note, *Brassicaceae*

have evolved the capacity to produce thioglucosides (also known as glucosinolates). These are also indole-type phytoalexins[15,33], however, although direct transport evidence for ABCG36 is lacking. Moreover, both IBA and CLX are tryptophane derivates that are as such metabolically connected[18,33]. Based on such a chemical overlap between transported defense and growth substances, we proposed that the extracellular gate might have gained a special role as a selectivity filter whose composition may have shifted according to the unique metabolite/defensive compound profile in the *Brassicaceae* family.

The extracellular gate of ABCGs contains the sequence "Gly-hydrophobic-hydrophobic" in both halves of human ABCG2[26] or yeast Pdr5[49] (Supplementary Fig. 7a). To get a deeper insight into the plant evolution of the key residues of the extracellular gate, we employed a recently published library of 1853 plant, full-size ABCG transporters extracted from the 1KP project[29,50,51]. With very few exceptions, the glycine is strictly conserved in all other Arabidopsis and plant ABCGs (Fig. 6; Supplementary Fig. 7a). As for human ABCGs, the "leucine valve" itself is only poorly conserved in Arabidopsis and other plant ABCGs (Fig. 6; Supplementary Fig. 7). The first leucine following the conserved glycine is in most cases substituted by a phenylalanine (F703 in ABCG36), with an overall **G F** x x S/P R/K x x consensus motif (Fig. 6a). Overall, the residues of TH5 contributing extracellular gate are slightly less conserved than that of TH11 (Fig. 6a, b).

We emphasized our analyses on the conservation of ABCG36 residues F703/L704 and F1374/F1375 as they are crucial residues of the extracellular gate limiting the translocation channel and are in close contact to the bound substrates, IBA and CLX (Fig. 6c;[26,30,31]). We hypothesized that variations in the recognition of diverse substrates in various plant ABCG proteins might be reflected by their degree of conservation. We found that L704 and F1375 that are located most centrally in the translocation chamber (Supplementary Fig. 7b), revealed a far lower degree of conservation than F703 and F1374 (Fig. 6c; Supplementary Fig. 7b). While for F703 and F1374 of Arabidopsis ABCG36 in most cases (>90%) the aromatic but hydrophobic phenylalanine is preserved, in other plant species L704 and F1375 are substituted by a limited choice of non-aromatic, hydrophobic amino acids, predominantly isoleucine, leucine and valine. Interestingly, in direct comparison, F1375 reveals a slightly higher level of variation than L704, the former also employing methionine and threonine (Fig. 6a–c).

Since higher plants have an advanced degree of chemodiversity provided by their specialized metabolism allowing for a sophisticated lifestyle, including advanced development and defense reactions[8], we compared the conservation of these key residues in the *Brassicaceae*, *Fabaceae*, and *Solanaceae* families containing beside Arabidopsis a series of important crop plants, like cabbage variations, soybean and tomato. While the overall degree of high and low conservation for F703/ F1374 and L704/ F1375, respectively, did not greatly differ amongst the families (Fig. 6d), we found a significantly higher and more equally distributed variability for L704 in the *Brassicaceae* family

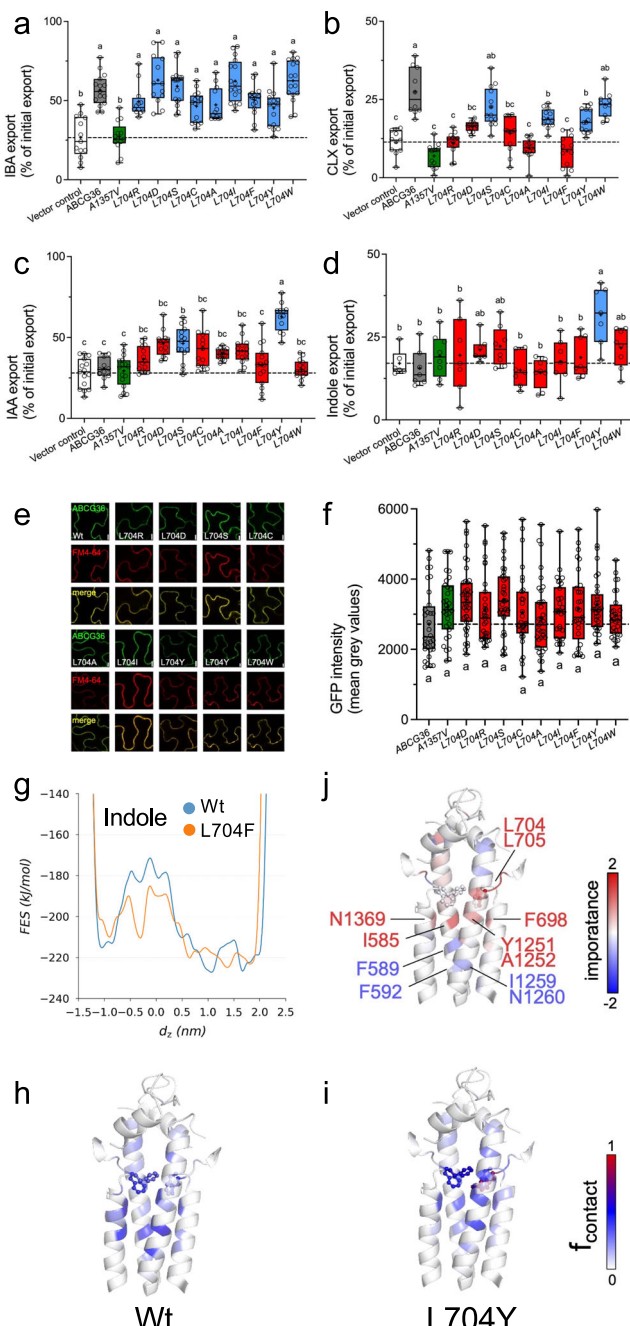

**Fig. 5 | L704Y mutation in ABCG36 widens the transport selectivity to indolic compounds.** IBA (**a**) CLX (**b**), IAA (**c**), and indole (**d**) export from *N. benthamiana* protoplasts after transfection with indicated mutant versions *ABCG36*. Means of mutant ABCG36 that are significantly different from Wt ABCG36 (grey fill) are indicated in red, while non-significant ones are in blue. ABCG36[A1357V] (green) is included as a negative control. Significant differences ($p < 0.05$) of means ± SE ($n = 14$ independent protoplast preparations) were determined using Ordinary One-way ANOVA (Tukey's multiple comparison test) and are indicated by different lowercase letters. Confocal imaging (**e**) and quantification (**f**) of GFP-tagged versions of mutated *ABCG36* after transfection of *N. benthamiana* leaves. Short treatment of FM4-64 was used as PM markers; bar, 50 μm. Note that slightly enhanced PM signals of some mutant versions of ABCG36 in comparison to Wt are also found in respective FM4-64 controls and thus do not reflect higher expression levels; bar, 10 μm. Significant differences ($p < 0.05$) of means ± SE ($n = 25$ epidermal cells) were determined using Ordinary One-way ANOVA (Tukey's multiple comparison test) and are indicated by different lowercase letters. **g** Free energy surfaces (FES) were calculated from metadynamics simulations with complexes of indole with Wt and L704Y ABCG36, respectively. **h–j** Pulling simulations of indole from the central binding pocket to the extracellular space were performed in simulations with Wt and L704Y ABCG36, respectively. Frequencies of contacts between the small molecule and the protein are plotted. **j.** A logistic regression model was employed to identify the key residues involved in defining transport. The model's coefficients were color-coded from blue to red, indicating residues whose interactions negatively (blue) or positively (red) support transport, respectively. Data are presented as box-and-whisker plots, where median and 25th and 75th percentiles are represented by the box itself and the middle line, respectively; means are indicated by a "+". Source data is provided as a Source Data file.

human ABCB1, ABCC1 and ABCG2, and yeast PDR5, were shown to significantly influence the affinity and function of drug binding[1,5,25,52–54]. Based on ABCG2 structures, residues L554 and L555 were suggested to form an extracellular "leucine plug", for ABCG-type transporters, conceptualizing an additional layer of substrate specificity control: the opening and closing of the 'leucine plug' may act as a checkpoint during the transport reaction. Khunweeraphong et al. (2019) extended this concept of a symmetric pump gate to six hydrophobic residues forming a "valve-like structure", contributing to the regulation of substrate efflux, potentially trading off translocation speed with substrate selectivity[22].

In this work we further hardened but also widely extended this overall concept by taking advantage of the recently published Arabidopsis *ABCG36* allele, *abcg36-5*, that contains a L704F point mutation in the extracellular gate and for which an uncoupling of ABCG36 functions in IBA-stimulated root growth from ABCG36 activity in extracellular defense was suggested[33]. Using heterologous and homologous transport systems (Figs. 1, 2) we clearly demonstrate that this uncoupling of ABCG36 function in growth from defense is due to a unilateral loss-of CLX transport capacity, while IBA transport was preserved. This contrasts with the *abcg36-6* allele that contains an A1357V substitution in the residue, which is a part of TH11, but not part of the extracellular gate. Importantly, in contrast to the T-DNA insertion mutant (null) allele, *abcg36-4*, used here as a negative control, both ABCG36[L704F] and ABCG36[A1357V] are expressed at similar levels on the PM (Figs. 1g, h and 2e, f,[33]).

ATPase activities (Fig. 1i) of ABCG36[L704F] were comparable to Wt ABCG36 and showed likewise a stimulation by IBA and CLX, while ABCG36[A1357V] were slightly enhanced but lacked substrate stimulation. This indicates that unlike ABCG36[L704F] mutation in ABCG36[A1357V] resulted most likely in an uncoupled, transport-incompetent transporter. Interestingly, in contrast to L555A exchange in ABCG2 resulting in lowered substrate-stimulated ATPase activity but twice the translocation activity of the wild-type[22], the L704F mutation in ABCG36 preserved ATPase activity and unilateral transport capacity.

It is still remarkable that the mutation of L704 leads to a total disruption of CLX transport, while IBA transport is entirely preserved.

with a shift toward leucine and valine on the cost of phenylalanine. A similar shift toward leucine was also found for F1375.

Together with our transport and molecular dynamics data, these data support the evolution of L704 (and F1375) as *Brassicaceae* family-specific key residues of the extracellular gate that controls the identity of chemically similar substrates during their transit from the substrate-binding cavity to the upper cavity. Such an evolutionary shift was apparently necessary in the *Brassicaceae* family because ABCG36 is confronted with a series of chemically very similar but physiologically very distinct substrates.

## Discussion

Despite the growing number of reported ABC transporter substrates and structures, the mechanisms underlying substrate identification and differentiation remain widely unclear. Intensive mutational analyses of TH residues that are thought to form substrate-binding sites and translocation pathways of paradigm ABC transporters, including

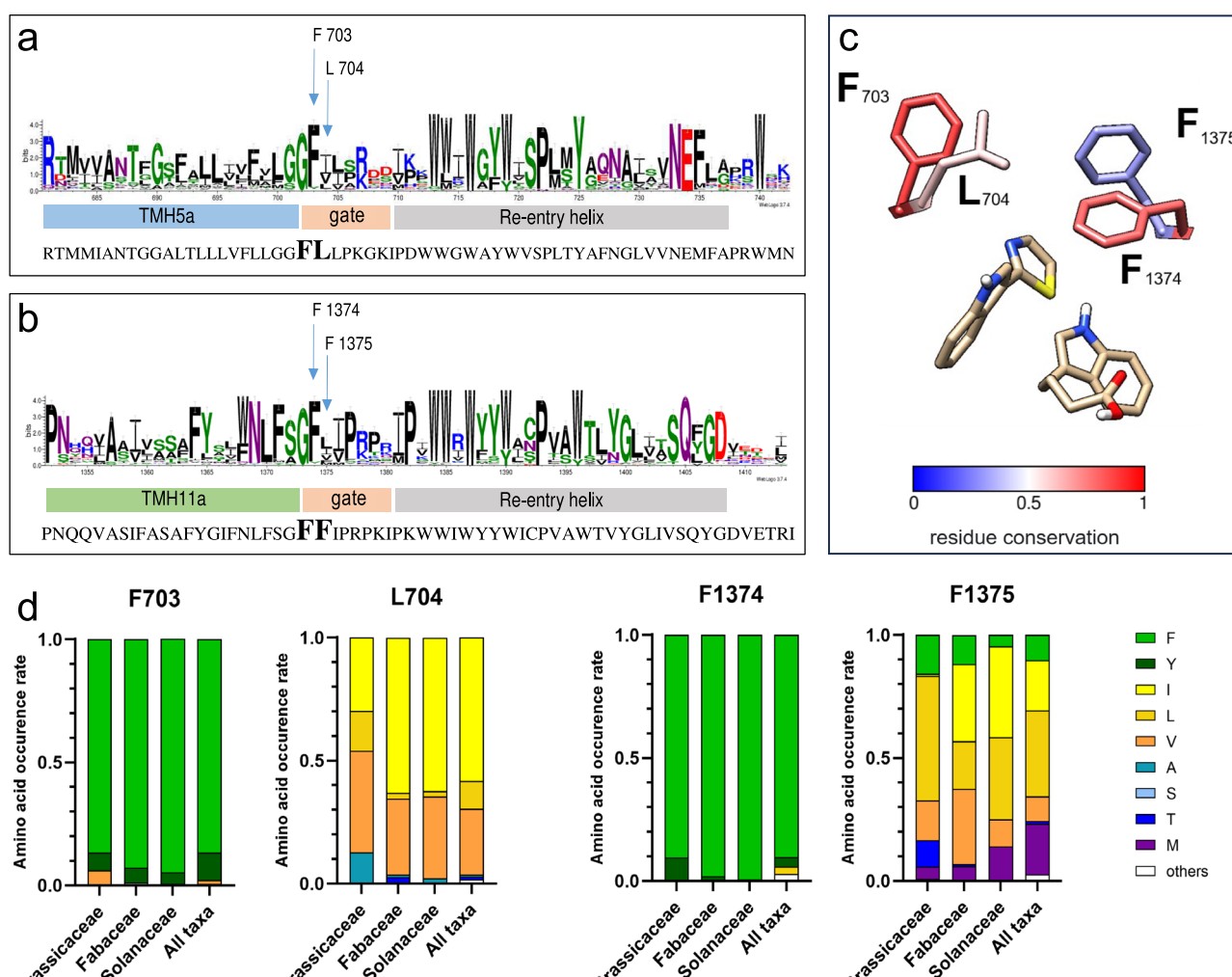

**Fig. 6 | Evolutionary analysis of key residues in the putative extracellular gate.** WebLogo presentation of the extracellular gate regions of TH5 (**a**) and TH11 (**b**) of 1853 full-size ABCGs. The x-axis displays the position of amino acids in the multiple sequence alignments according to AtABCG36. **c** L704 and F1375 are less conserved than F703 and F1374 in full-size ABCG sequences, the positions of ABCG substrates, IBA, and CLX are indicated. **d** Occurrence of amino acids corresponding to F703, L704, F1374, and F1375 in Arabidopsis ABCG36 in full-size ABCG sequences of indicated taxa.

However, this finding on the transport level is absolutely in agreement with the published and here shown growth phenotypes of the *pen3-5/L704F* mutant (Supplementary Fig. 1)[33]. Moreover, indirect support comes from work on Medicago ABCG46 (belonging to the *Fabaceae*), where F562L (corresponding to F431 in HsABCG2) leads to selective loss of transport of the phenylpropanoid, 4-coumarate, while liquiritigenin is still a functional substrate[29].

Molecular dynamics simulations of WT and L704F mutant versions of ABCG36 (Fig. 4), based on an AlphaFold2-predicted structures in a closed conformation, provided metadynamics and simulation-based rational for the transport data. These simulations highlighted molecular determinants explaining why IBA is a substrate for both WT and L704F ABCG36, while CLX is only transported by WT ABCG36 (Figs. 1, 2). Differences in the FES profiles around the central binding pocket and interactions of molecules as they are pulled toward the extracellular space (Fig. 4) suggest that substrate export is governed not solely by the valve-region but distinct regions of the transport pathway. For instance, metadynamics simulations with the non-substrate IAA revealed high energy barriers for IAA movement around the central pocket, explaining ABCG36's remarkable ability to discriminate between the structurally similar IBA and IAA (Supplementary Fig. 5a, b). DFT analyses provided evidence that this high

barrier for IAA might be caused by drastic deformation of the electronic structure after binding to the ABCG36 binding pocket (Fig. 4c, d). This change in the electron system results in a positive binding energy preventing transport. This capability of ABCG36 is apparently essential because the IAA precursor IBA is thought to function in overlapping but also distinct developmental pathways[18,48].

Our simulations with IBA and CLX indicate that multiple mechanisms underlie substrate recognition, including entry into the central binding pocket, interactions within the pocket, and interactions near the extracellular gate. Logistic regression analysis (Fig. 5i) exemplified key residues influencing transport: interactions with residues including I585, F698, L704, L705, Y1251, A1252, and N1369 were associated with promoting transport, while contacts with residues like F589, F592, I1259, and N1260 counteracted the process. These results highlight an asymmetry in the substrate recognition pattern, potentially linked to the asymmetry in ATP hydrolyzing capacity of the NBDs. Furthermore, this asymmetry is evident at the extracellular gate, where direct interactions of L704 with substrates are crucial for transport, as indicated by high logistic regression coefficients (Fig. 5j). Importantly, we hypothesize that the observed pattern is highly adaptable and may change depending on the molecule or mutation. Identifying the shared mechanism of allosteric signaling underlying this plasticity is essential

for understanding and predicting substrate interactions with multi-drug transporters.

In summary, mutations in the extracellular gate likely impact small molecule interactions at two distinct levels. Beyond the direct interactions between key valve residues and small molecules, our in silico findings highlight the critical role of allosteric interactions between the valve residues and the central binding pocket in shaping substrate recognition and transport.

Mutational work further supported the conclusion that L704 is a key residue of the extracellular gate that provides quality control, contributing to substrate specificity of ABCG36 toward a few indolic compounds. All tested mutations of L704 (including L704F and L704A) preserved IBA transport capacities, while this was not the case for CLX export (Fig. 5a). This might suggest that the additional aromatic ring of CLX allows for discrimination by using a contact with L704. Remarkably, exchange of L704 against the aromatic residue, tyrosine, but not for structurally similar tryptophane or phenylalanine resulted in a gain-of-IAA and indole transport (Fig. 5). Competition experiments excluded that L704Y mutation did result in an uncoupling of ATPase activity and transport leading to a permanently open conformation and thus an unspecific transporter (Supplementary Fig. 2).

This broadening of substrate specificity by L704Y but not L704F or L704W mutations is remarkable because all three aromatic residues are chemically and structurally similar; e.g. phenylalanine and tyrosine amino acids differ just by a hydroxyl group. Therefore, we conclude that the change of a leucine to the aromatic phenylalanine limited transfer through the extracellular gate, while further addition of a hydroxyl group widened substrate specificity. Interestingly, tyrosine is absent in any taxa of higher plants for the L704 position, but it is frequent for positions of F703 and 1374 (Fig. 6) and is part of the so-called "apoplastic gate" in the ABA exporter ABCG25 (Y565;[30]).

Taken together, our transport, metadynamics simulation and evolutionary work support the evolution of L704 as a *Brassicaceae* family-specific key residue of the extracellular gate that may control the translocation process in different regions, allosterically at distant parts based on simulations and directly indicated by experiments. Initially, we aimed to correlate the composition of the extracellular gate with known substrate specificities of plant PDRs (including IBA, CLX, and ABA), but this trial failed. Also, data from metadynamics simulations did not reveal large differences in FES around the extracellular gate of various compound/protein complexes, except in the case of IBA. As such, we believe that the gate does not exclusively contribute to substrate specificity but that it instead may provide a final quality check of chemically closely related molecules, especially for the case that a non-substrate was falsely recognized in the central binding pocket region.

From the plant's perspective, such a final quality control on the substrate level provided by the extracellular gate of ABCG36 is of major importance because, as mentioned above, both ABCG36 substrates, IBA and CLX, serve as crucial substrates in two separate programs, growth and defense, respectively. While the need for such a final quality control is obvious for targeted CLX export provided by ABCG36 expressed toward the attacking pathogen[39], the opposite scenario might also be critical. A physiological consequence of prolonged defense is growth inhibition[16,17] because essential resources are thought to be allocated toward defense[16]. As mentioned before, camalexin and IBA are also metabolically connected[18,48], making such a strict quality control also necessary from a metabolic point of view. On the molecular level, this coordination of substrate preferences on ABCG36 was shown to be provided by ABCG36 phosphorylation of the linker (mainly S825 and S844) by QSK1 repressing IBA export and allowing for CLX export and thus defense[14]. Currently, it is unknown if these phosphorylation events exert a long-range effect on the central binding pocket or on the extracellular gate. However, both options and their combination are viable, since the existence of

this type of regulation and our findings suggest that distant regions of ABCG transporters are likely to exhibit such a strong allosteric coupling as ABCC proteins[55]. A puzzling question is how phosphorylation of a single serine on the obvious flexible and disordered regions of ABCG36, can have such a strong effect on ABCG36 substrate specificity. One explanation might be that these regions function in analogy to the regulatory linker or R-domains of CFTR/ABCC7 or YEAST CADMIUM FACTOR (YCF1)[56]. For yeast PDRs/ABCGs, it was shown that NBDs influence transport activity but also substrate specificity[57]. Such a long-range effect is eventually not caused <u>directly</u> by phosphorylation of these flexible loops but <u>indirectly</u> by allowing interacting, regulatory proteins to bind to these surfaces[58].

In summary, our findings support the concept of a gate-keeper function of the extracellular gate and contribute to our understanding of how ABCG substrate specificity is encoded. This is of interest because many ABCG proteins are also of clinical relevance, since their promiscuous substrate selectivity causes pleiotropic or multidrug resistance phenomena[1]. This is most obvious for the direct involvement of ABCGs in several genetic diseases (like ABCG5/G8 in sterol sitosterolemia or ABCG2 in gout) or chemotherapy failure (ABCG2) but also the treatment of fungal pathogens, like *Candida* species[1,59]. Infections by Candida, themselves containing a plethora of ABCG transporters, are increasingly providing a threat to immunosuppressed people[59].

An evolutionary analysis uncovered that, like for human ABCGs, the extracellular gate is poorly conserved in Arabidopsis and other plant ABCGs (Fig. 6; Supplementary Fig. 6). L704 and F1375, which are in closest vicinity to the transported substrates and that are located most centrally in the translocation chamber (Fig. 4c; Supplementary Fig. 7b) revealed a far lower degree of conservation than F703 and F1374 (Fig. 4c). A comparison of residue conservation in the major crop families uncovered, in contrast to the overall degree of high and low conservation for F703/ F1374 and L704/ F1375, respectively, which does not greatly differ amongst the families (Fig. 6d), a more equal distribution and variability for L704 in the *Brassicaceae* family but not for F1375. A reasonable explanation is that an enhanced variability at this position has allowed ABCG36-like transporters in the *Brassicaceae* family to accommodate several chemically similar indolic substrates (here: CLX and IBA[13]) but as well other indolic compounds functioning in defense signaling[15] that are either crucial for development and defense, respectively. It appears that in the *Brassicaceae* family ABCGs are confronted with chemically very similar but physiologically very distinct substrates, requiring a special level of quality control provided by the extracellular gate (most obvious for L704), which as a result has evolutionary separated.

## Methods
### Plant material
The *following Arabidopsis thaliana* lines in ecotype Columbia (Col Wt) were used: *abcg36-4/pen3-4* (SALK_000578, Stein *et al.* 2006), *abcg36-6/pen3-6/pdr8-115* (Strader *et al.* 2009). gl1 (gl1 Wt) was used as the wild-type control for *abcg36-5/pen3-5*[33]. Mutant versions of *ABCG36* were obtained by QuickChange (Agilent Company, USA) site-directed mutagenesis of *35S:ABCG36* (Kim *et al.* 2007), *35S:ABCG36-GFP*[14] and *ABCG36:ABCG36-GFP*[39]. Forward (FW and revers (RV) primers were as follows: L704R_FW: 5'-TTC CCC TTA GGG AGT CTA AAA CCA CCC AAA AGA AAT ACC AAC AAC A-3'; L704R_RV: 5'-TGT TGT TGG TAT TTC TTT TGG GTG GTT TTA GAC TCC CTA AGG GGA A-3'; L704D_FW: 5'-ATT TTC CCC TTA GGG AGA TCA AAA CCA CCC AAA AGA AAT ACC AAC AAC AAC-3'; L704D_RV: 5'-GTT GTT GTT GGT ATT TCT TTT GGG TGG TTT TGA TCT CCC TAA GGG GAA AAT-3'; L704S_FW: 5'-TTC CCC TTA GGG AGT GAA AAA CCA CCC AAA AGA AAT ACC AAC AAC A-3'; L704S_RV: 5'-TGT TGT TGG TAT TTC TTT TGG GTG GTT TTT CAC TCC CTA AGG GGA A-3'; L704C_FW: 5'-GAT TTT CCC CTT AGG GAG GCA AAA ACC ACC CAA AAG AAA TAC CAA CAA CAA CGT-3'; L704C_RV: 5'-

ACG TTG TTG TTG GTA TTT CTT TTG GGT GGT TTT TGC CTC CCT AAG GGG AAA ATC-3'; L704A_FW: 5'-CCC CTT AGG GAG TGC AAA ACC ACC CAA AAG AAA TAC CAA C-3'; L704A_RV: 5'-GTT GGT ATT TCT TTT GGG TGG TTT TGC ACT CCC TAA GGG G-3'; L704I_FW: 5'-CCC CTT AGG GAG TAT AAA ACC ACC CAA AAG AAA TAC CA-3'; L704I_RV: 5'-TGG TAT TTC TTT TGG GTG GTT TTA TAC TCC CTA GGG GG-3'; L704Y_FW: 5'-GAT TTT CCC CTT AGG GAG ATA AAA ACC ACC CAA AAG AAA TAC CAA CAA CAA CGT-3'; L704Y_RV: 5'-ACG TTG TTG TTG GTA TTT CTT TTG GGT GGT TTT TAT CTC CCT AAG GGG AAA ATC-3'; L704W_FW: 5'-GAT TTT CCC CTT AGG GAG CCA AAA ACC ACC CAA AAG AAA TAC CAA CAA CAA CGT-3'; L704W_RV: 5'-ACG TTG TTG TTG GTA TTT CTT TTG GGT GGT TTT TGG CTC CCT AAG GGG AAA ATC-3'. Mutated plasmids were used to transform *abcg36-4* by floral dipping to generate *ABCG36:ABCG36^{L704F}-GFP* (*abcg36-4*) #1, *ABCG36:ABCG36^{L704F}*-GFP (*abcg36-4*) #2, *ABCG36:ABCG36^{A1357V}-GFP* (*abcg36-4*) #1, *ABCG36:ABCG36^{A1357V}*-GFP (*abcg36-4*) #2, *ABCG36:ABCG36^{L704F A1357V}*-GFP (*abcg36-4*) #1 and *ABCG36:ABCG36^{L704F A1357V}*-GFP (*abcg36-4*) #2. Isogenic, homozygous lines for the transgene in the F3 generations were used for further analyses.

## Plant growth conditions

Seedlings were generally grown on vertical plates containing 0.5 Murashige and Skoog media, and 0.75% phytoagar at 12 h light per day. Developmental parameters, such as primary root lengths, were quantified using standard methods[40]. All experiments were performed at least in triplicate with 30 to 40 seedlings per experiment.

## *Fusarium oxysporum* elicitor treatments and root infection assays

*Fusarium oxysporum f. sp. conglutinans* (Fo), strain 699 (Fo699; ATCC 58110), and strain 5176 (Fo5176) was used throughout this study, and a fungal elicitor mix was prepared from Fo5176[60]. *F. oxysporum* infection assays of Arabidopsis roots in soil were performed with strain Fo699 as reported in ref. 61. In short, 3-week-old *Arabidopsis thaliana* plants were infected by pipetting 10 ml conidia suspension ($10^7$ conidia/ml) directly into the soil contained in a 125 ml plastic pot harboring a single plant. Fresh weight and disease sensitivity scores were obtained by measuring rosette weight and by observing chlorotic and necrotic symptoms scored at a scale from 1-10 as a number of affected leaves per plant, respectively, two weeks after inoculation. 15-30 plants were employed per genotype per experiment in each of three independent experiments (n = 3).

## *Phytophthora infestans* infection assays

Rosette leaves of 5-week-old *A. thaliana* plants were drop-inoculated with zoospore suspensions of *P. infestans* Cra208m2[62] or of *B. cinerea* B 05.10 by applying 10 μl drops containing 5 x $10^5$ or 5 x $10^4$ spores ml$^{-1}$, respectively, onto the adaxial leaf surface. Thirty drops each were collected and immediately frozen in liquid nitrogen for storage. For extraction, the drops were evaporated to dryness, and samples were dissolved in 60 μl of 30% methanol, sonicated for 15 min, and analyzed by LC-MS.

Inoculation of seedlings was performed as follows[34]: Arabidopsis seedlings (Col-0, *abcg36-6*, Gl1 Wt and *abcg36-5*) were grown in 2 ml liquid medium (1/2 strength MS with sucrose) for 12 days. One day before inoculation with *B. cinerea* (5 x $10^3$ ml$^{-1}$) or *P. infestans* zoospores (1 x $10^5$ ml$^{-1}$), the medium was removed and replaced with 2 ml 1/2 MS without sucrose. 600 μl of the medium were collected 24 h post-inoculation, evaporated to dryness, and re-dissolved in 120 μl of 30% methanol. CLX levels were determined by UPLS-ESI-QTOF-MS.

## Transport and binding studies

$^3$H-Indole (ARC ART2269, 25 Ci/mmol), $^3$H-IAA (ARC ART0340, 25 Ci/mmol), $^3$H-IBA (ARC ART2533, 10 Ci/mmol), $^{14}$C-BA (ARC ART0186A, 55 mCi/mmol) and $^3$H-CLX (3-thiazol-2-yl-indole; custom-synthesized

by ARC, 56 Ci/mmol) export from *Arabidopsis* and *Nicotiana benthamiana* mesophyll protoplasts was analyzed as follows[41]: tobacco mesophyll protoplasts were prepared 4 days after Agrobacterium-mediated transfection. Equal protoplast loading was achieved by substrate diffusion into protoplasts on ice and export was started by a temperature shift (25 °C). Exported radioactivity was determined by separating protoplasts and supernatants by silicon oil centrifugation. Relative export from protoplasts is calculated from exported radioactivity into the supernatant as follows: (radioactivity in the supernatant at time $t$ = x min.) - (radioactivity in the supernatant at time $t$ = 0)) * (100%)/ (radioactivity in the supernatant at $t$ = 0 min.). In some cases, relative import into tobacco protoplasts was determined by separating protoplasts and supernatants by silicon oil centrifugation. Loading (uptake) was calculated from imported radioactivity into protoplasts as follows: (radioactivity in the protoplasts at time t = x min.) - (radioactivity in the protoplasts at time $t$ = 0)) * (100%)/ (radioactivity in the protoplasts at $t$ = 0 min.). Presented are mean values from at least 4 independent transfections and/or protoplasts preparations at $t$ = 15 min.).

$^3$H-IAA and $^3$H-Indole uptake into *Arabidopsis* vesicles prepared from Arabidopsis lines grown as mixotrophic liquid cultures was measured in the absence (solvent) or presence of 1000 x access of nonlabelled Indole, IAA, IBA, or CLX, respectively[15].

Drug binding assays using Arabidopsis microsomes were performed as follows[63]: four replicates of 20 μg protein each were incubated with 10 nM radiolabeled $^3$H-IBA or $^3$H-CLX in the presence and absence of 10 μM non-radiolabeled IBA or CLX, respectively. Reported values are the means of specific radiolabeled drug bound in the absence of cold drug (total) minus radiolabeled drug bound in the presence of a 1000-fold excess of cold drug (unspecific) from at least 4 independent experiments (independent microsome preparations) with four replicates each.

## Measurement of ATPase activity

ATPase activity was measured from microsomes (0.06 mg/ml) prepared from tobacco plants transfected[41] with vector control or Wt and mutant versions of *35S:ABCG36-GFP* using the colorimetric determination of ortho-phosphate released from ATP[64]. Briefly, microsomes were added to ATPase buffer (20 mm MOPs pH 7.7 or pH 9.0, 8 mm MgSO$_4$, 50 mm KNO$_3$, 5 mm NaN$_3$, 0.25 mm Na$_2$MoO$_4$, 2.5 mm phosphoenolpyruvate, 0.1% pyruvate kinase) in the presence and absence of 0.5 mM sodium ortho-vanadate. The reaction was started by the addition of 15 mm ATP and incubated at 37 °C for 15 min with shaking. The amount of Pi released in the absence (solvent control) or presence of IBA, IAA, CLX, indole, or *ortho*-vanadate (50 μM) was quantified using a Cytation 5 reader (BioTek Instruments).

## Metabolite profiling using *LC-MS*

Untargeted metabolite profiling[15] and targeted relative quantification of metabolites were done on the peak areas of characteristic extracted ion chromatograms in Bruker's QuantAnalysis V4.1 software (Bruker Daltonik).

## Confocal laser scanning

For imaging, seedlings were generally grown for 5 days on vertical plates containing 0.5 Murashige and Skoog media, 1% sucrose, and 0.75% phytoagar at 16 h (long day) light per day. For chemical treatments, seedlings were transferred for 12 h on test plates containing the indicated chemicals or solvents. For confocal laser scanning microscopy work, an SP5 confocal laser microscope was used. Confocal settings were set to record the emission of GFP (excitation 488 nm, emission 500-550 nm), and FM4-64 (excitation 543 nm, emission 580-640 nm). Images were quantified with ImageJ (http://imagej.nih.gov/ij/) using the Fiji plugin and identical settings for all samples. As regions of interest either whole images or individual cells

marked using the freeform tool were used for quantification; these are indicated in the legends.

## Protein structures and chemical formulas

The AlphaFold2 predicted ABCG36 structure (AF-Q9XIE2-F1) was[65,66]. ATP molecules and $Mg^{2+}$ ions were inserted after the structural alignment of nucleotide-binding domains of this ABCG36 structure and our formal ABCG2 homology model[67]. Regions with high uncertainty (residues 1–39 and 809–863) were deleted since they may exert unwanted effects in molecular dynamics simulations. IBA, IAA, IND, and CLX 3D structures were downloaded from PubChem (IDs: 8617, 802, 798, and 636970).

## In silico docking

The small molecules were docked to the extracellular (EC) side of the extracellular gate of the AF-predicted ABCG36 structure using AutoDock Vina[68]. The following options were set: grid size = 22.50 Å, center_x = 1.4, center_y = 0.22, center_z = 15, exhaustiveness = 64. The predicted pose with the lowest score/binding energy was used in MD simulations.

## Molecular dynamics simulations

The input files for all steps (energy minimization, equilibration, and production run) were generated by the CHARMM-GUI web interface[69] by submitting the structure of a complex, including the protein, ATP, and a small molecule. CHARMM36m all-atom force field was used[70] and IBA, CLX, IAA, and indole were parametrized by CHARMM General Force Field (CGenFF)[71]. Periodic box conditions were applied with a rectangular box. The full-length structures were oriented according to the OPM database. Mutations were introduced via CHARMM-GUI. A mixed membrane bilayer was built, including 1:1 POPC: PLPC (1-palmitoyl-2-oleoyl-sn-glycerol-3-phosphocholine:1-pal-methyl-2-linoleoyl-sn-glycerol-3phosphocholine) in the extracellular leaflet and 37:43:12:8 POPC: PLPC: POPS: DMPI25 (POPS: 1-palmitoyl-2-oleoyl-sn-gglycerol3-phospho-L-serine, DMPI25: dimyristoyl-phosphatidylinositol 2,5) in the intracellular leaflet. The following additional options were adjusted: i) terminal residues were patched by ACE (acetylated N-terminus) and CT3 (N-methylamide C-terminus), ii) 150 mM KCl in TIP3 water was used, iii) grid information for PME (Particle-Mesh Ewald) electrostatics was generated automatically and iv) a temperature of 303.5 K was set. The structures were energy minimized using the steepest descent integrator (maximum number to integrate: 50,000 or converged when force is <1,000 kJ/mol/nm). From the energy-minimized structures, an equilibrium simulation was forked for 100 ns. During these simulations, a Berendsen thermostat and a Parinello-Rahman were applied for equilibration and production runs, respectively. Besides, Parinello-Rahman barostat with a semi-isotropic, 1 atm pressure coupling, a time constant of 5 ps in the plane of the membrane, with the compressibility of 4.5 x $10^{-5}$ $bar^{-1}$, was applied[71,72]. The cut-off value considered for the long-distance interactions was 12 Å. Electrostatic interactions were calculated using the fast, smooth PME algorithm, and the LINCS algorithm was used to constrain bonds at 1.2 nm. Constant particle number, pressure, and temperature ensembles were used with a time step of 2 fs. Simulations were performed using GROMACS 2022. Well-tempered metadynamics simulations were executed with GROMACS 2022 patched with PLUMED 2.8.1[73]. Multiple walker setups (n = 4 or 8, depending on hardware) were used. The total simulation time was 2 μs for each complex (total time: 7 systems x 2 μs = 14 μs). CV was defined as the Z component of the distance between the center of geometry of ABCG36 residues in the central transmembrane helices (a.a 582-597: FGMIINMFNGFAEMAM, 685-700: IANTGGALTLLLVFLL, 1250:1265: LYAAIIFVGINNCSTV, 1356-1371: VASIFASAFYGIFNLF). The parameters used were ARG = CV PACE = 500 (1 ps), HEIGHT = 0.6, SIGMA = 0.05, BIASFACTOR = 10.0, TEMP = 303.15 K, GRID_MIN = -1.5 GRID_MAX = 2.4, GRID_WSTRIDE = 1000. The movements were constrained with

LOWER_WALL (dz = -1.1 nm) and UPPER_WALL (dz = 2.0 nm) to inhibit the escape of the small molecules from the four helices that would have rendered the simulations unfeasible due to the extremely low probability of returning to the pathway. FES was calculated using the sum_hills tool of PLUMED.

For simulations pulling each compound from the intracellular to the extracellular side of the extracellular gate, the system was prepared as follows. First, PLUMED was used to pull the small molecule into the binding pocket, using the same CV as for metadynamics simulations. The parameters used were ARG = CV, STEP0 = 0, AT0 = INITIAL_DISTANCE, KAPPA0 = 1000, STEP1 = 4,500,000, AT1 = 0.2, STEP2 = 5,000,000, AT2 = 0.2, KAPPA2 = 0. Then, a short (10 ns) constrained simulation was performed to keep the small molecule at the position of 0.2. The resulting trajectory was sampled for 25 frames with different velocities as input for pulling simulations. The pull-out was performed using the GROMACS pull code with umbrella potential. Small molecules were pulled along the positive Z axis (Supplementary Fig. 4c) with a force constant and velocity of 500 kJ/mol/nm and 0.0005 nm/ns respectively. Contact frequencies were calculated and adjusted based on the awakening forces within ±10 ps around each frame. The correction factor was defined as correction_factor = (positive_forces − |negative_forces|) / (positive_forces + |negative_forces|).

Analysis was done using MDAnalysis[74] and Matplotlib. Python packages. PyMOL Molecular Graphics System (Version 2.0 Schrödinger, LLC) was used for molecular visualization tasks.

## Quantum chemical modeling

The initial ABCG36 geometry[65,66] was derived from protein structure predictions using AlphaFold2 (AF-Q9XIE2-F1). The size of the binding pocket region was determined through docking methods using AutoDock Vina. Molecular dynamics were used to perform several geometric conjugations of a pocket of 630 atoms with each of the different ligands, using MM+ force fields and their respective pre-optimizations. At least two conformations pre-optimized at the MM+ level were taken for each ligand.

The following step involved obtaining optimizations using Density Function Theory methods, which included a Van der Waals D3 correction (DFT-D) with the bp86 functional and the double zeta polarized atomic basis set level [def -SV(P)]. After the calculation of each molecular component, the pocket-ligand binding energies were determined by first calculating the electronic interaction energy using the equation $E_{el} = E_{PL} - (E_P + E_L)$, where $E_{el}$ is the electronic binding energy, $E_{PL}$ is the electronic interaction energy between the pocket and ligand, $E_P$ is the electronic interaction energy within the pocket, and $E_L$ is the electronic interaction energy of the ligand. All calculations were performed with Turbomole (http://www.turbomole.com). The analysis of the deformation of the electronic structure and changes in the potential surface was carried out using the theory of deformation of atoms in molecules[75].

## Evolutionary ABCG analyses

To select full-size ABCG from multiple groups of plants, we subjected One Thousand Plant Transcriptomes (1KP) data (https://db.cngb.org/onekp/)[50,51] to tBLASTn searches using ABCG36 as a query sequence with an E-value cutoff of 1e−5. Obtained sequences were further filtered[29]. One thousand eight hundred and fifty-three 1KP samples were assigned to a full-size ABCG subfamily and used for the WebLogo analysis[76]. Multiple sequence alignment (MSA) of the full-size ABCG amino acid sequences was performed using the MUSCLE algorithm[77]. To analyze the amino acids occurrence rate in selected residues from the full-size ABCG sequences, firstly BLASTp search was performed independently for *Brassicaceae, Fabaceae*, and *Solanaceae*, using the sequence of AtABCG36 as a query with an E-valu cut-off of 0.05. Obtained sequences were then filtered by length (between 950 and 1800) and checked to contain plant PDR signature motives[78] (LLXGPP,

GLXSS, GLDARXAAXVMR, VCTIHQP). 598 sequences from *Brassicaceae*, 762 from *Fabaceae*, 450 from *Solanaceae*, and 1599 sequences from all taxa (obtained from 1KP) were aligned using the MUSCLE algorithm and analyzed using Python.

## Statistics & reproducibility

Data were statistically analyzed using Prism 10.1.1. (GraphPad Software, San Diego, CA). Normal (Gaussian) distribution of values was tested prior to statistical analyses using the Prism-embedded D'Agostino-Pearson omnibus normality test. For statistics either "Ordinary One-way ANOVA" or "Two-way ANOVA" and "Tukey's multiple comparison test" was used as post-hoc analysis to determine significances. Data are presented as "box-and-whisker plots", where median and 25th and 75th percentiles are represented by the box itself and the middle line; means are indicated by a "+".

No statistical method was used to predetermine sample size. No data were excluded from the analyses; the experiments were not randomized; the Investigators were not blinded to allocation during experiments and outcome assessment.

## Reporting summary

Further information on research design is available in the Nature Portfolio Reporting Summary linked to this article.

## Data availability

This article does not contain any original code. Requests for data should be made to and will be fulfilled by M.M. Geisler (markus.geisler@unifr.ch), provided the data will be used within the scope of the originally provided informed consent. Source data are provided with this paper. Important input and output files of MD simulations are available at Zenodo (https://doi.org/10.5281/zenodo.14743311). Source data are provided with this paper.

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

## Acknowledgements

We would like to thank L. Charrier (UNIFR) and U. Smolka (IPB) for excellent technical assistance, the computational resources made available on the GPU cluster Komondor (Governmental Information-Technology Development Agency, Hungary), and T. Romeis (IPB) and J. Huyhn (UNIFR) for experimental support. This work was supported by grants from the Swiss National Funds (project 31003A_165877 and 310030_197563 to MG), the *Pool de Recherche* of the University of Fribourg (to MG), the Polish National Science Center (grant 2017/27/B/NZ1/01090 to MJ and 2021/41/N/NZ1/04030 to MJ and KP), the German Research Foundation (grant RO 1172/6–1 to SR), the National Research, Development and Innovation Office (grant NKFIH-137610, 2024-1.2.3-HU-RIZONT-2024-00003 and TKP2021-EGA-23 to TH) and the China Scholarship Council (to JX).

## Author contributions

M.M.G., T.H. and J.X. designed research; J.X., A.S., F.I., O.T., J.B., K.P., J.Z., N.F., and T.H. performed research; J.X., A.S., F.I., O.T., J.B., K.P., J.Z., S.R., N.F., T.H., and M.M.G. analyzed data; M.J., T.H. and M.M.G. supervised work; M.M.G. and T.H. wrote the manuscript, all authors commented on the manuscript.

## Competing interests

The authors declare no competing interests.
