## [Transparent Peer Review file · Nature Communications]

A key residue of the extracellular gate provides quality control contributing to ABCG substrate specificity

Corresponding Author: Dr Markus Geisler

Version 0:

Reviewer comments:

Reviewer #1

(Remarks to the Author)

The manuscript by Xia et al. presents a deep analysis of the substrate selectivity of the Arabidopsis ABC transporter, ABCG36, as governed by a key residue located along the transport path. Mutation of this residue from leucine to phenylalanine, L704F, reduces its camalexin (CLX) transport activity, while remains its IBA transport activity, which is supported by the extensive plant cell-based transport assay results carried out in the mutant background or the overexpression background. In silico simulations were performed to further support the conclusions. A broader mutagenesis of the key residue further found that the mutant L704Y expands the transport substrates, and allows transport of non-native ligands, including IAA and indole, further supporting that this residue plays essential roles in the substrate specificity of ABCG36.

Overall, the authors dissected the effect of the previously identified mutation in the substrate selection of this plant ABC transporter, and helped in understanding how such transporters transport multiple molecules with different chemical structures. Based on presented data, I believe the phenomenon observed and the conclusion reached by the authors are solid. I have several points and suggestions for the authors to consider to improve the quality of the manuscript, particularly regarding the MD simulation and molecular mechanism interpretation part.

Major points:

1. When carrying out the transport assays, the protein amount of ABCG36 in the PM should be carefully quantified. It would be difficult to judge the quantity of the WT protein and mutants simply from the confocal imaging results (Fig. 1g, etc.). Using another quantitative method would be much helpful to exclude the possibility that the observed transport differences were due to different protein levels.
2. The substrate export was calculated by measuring the relative radioactivity in the supernatant. How about directly measure the radioactivity in the cells, which would more faithfully reflect the export process? At least the authors should show this result for the representative control, WT, and L704F control. Since the authors did not describe the assay system in the method, it would be difficult to judge the change in the supernatant would be significant to reflect the changes. Also the meaning of the Y-axis labelling "% of initial export" is hard to understand, and should be replaced by another phrase.
3. In my opinion, the sequence conservation analysis presented in Fig.5 have limited meaning to understanding the substrate selectivity of ABCG transporters, particularly without the substrate information of the compared transporters. I suggest moving this part into the discussion and SI.
4. I appreciate the authors efforts in characterizing the transport and ATPase activity of the ABCG36 constructs. I wonder if the authors could manage to measure the binding affinity between ABCG36 and the different substrates. It would help to understand how the mutation poses effects to the transport process and which step it might affect.
5. The MD simulation for the transport process was carried by characterizing the pulling force of the ligand along the transport path. As for ABC transporters, the conformational changes would be the more direct process to exposing the substrate to different side of the PM. It would not be a good system to characterize the substrate selectivity using the pulling force method. Also the presented results for IBA and CLX in Fig.6f-j are not clear enough in my opinion to fully reveal the difference. It would be more interesting to have ABCG36 models under different transport conformations and compare the substrate binding profiles for different mutants, and whether L704 directly involved in the substrate binding under certain

states.

Minor points:

1. The x-axis labelling should be rearranged and show more clearly since they are almost overlapped, like in Fig.1h, Fig.3c-d, etc.
2. I noticed that the sample size presented in the transport assay result varied for different groups. It would be better to keep them in consistency and further improve the data quality.
3. The method part should provide more details of how the experiments are carried out, specially for the transport assay.

Reviewer #2

(Remarks to the Author)

The main substrates of the ABC transporter ABCG36 (PEN3 or PDR8) from *A. thaliana* are IBA (indole-3-butyric acid) and CLX (camalexin), respectively. IBA is a precursor of auxin, while CLX is an important antimicrobial compounds. As a member of the ABCG family, ABCG36 likely adopts the fold of human ABGG2. As this, it likely contains the a hydrophobic extracellular gate that controls substrate export. A mutant in ABCG36 (L704F) was already described that uncouples the transporter function. To explain this phenotype it was also suggested that CLX export was impaired while IBA export was similar to wildtype levels. In the submitted manuscript, the authors demonstrate that this is indeed the case. They elaborate this conclusion by different approaches including plant growth experiments, transport assays using protoplast, structural comparison of the ABCG2 structure and an AlphaFold derived model of ABCG36, and various simulation techniques. This is truly an important issue addressed by the authors and the data presented support the conclusion that IBA and CLX transport are uncoupled (IBA at wildtype levels and no CLX transport). I even have to say that I am impressed by the wealth of different methods applied. All these results come to the same conclusion. But my major criticism is the fact that a molecular explanation for such a phenotype is missing. Meaning that presented results remain descriptive in nature.

Below I list my points of concern in the order of appearance, not in the order of importance.

- (1) It is important to note that the extracellular gate contains the sequence Gly-hydrophobic-hydrophobic in both halves of human ABCG2 or yeast Pdr5. Is a Gly residue also present in ABCG36?
- (2) Line 182: I understand why ATPase measurements were performed at pH 9. But is this physiologically meaningful? Why not bench marking the activity against a sample that contains an inactive ABCG36 version?
- (3) Line 307: Here a first glimpse of a molecular explain for the abrogation of CLX transport is provided – a stronger interaction of CLX with the transporter. If this is indeed true, what happens in mutants that reduce these stronger interactions to the IBA situation?
- (4) Section sequence conservation. If the observed differences in conservation and in the different families has a physiological implication, is it possible to generate a transporter with a similar phenotype as ABCG36 (one substrate transported, the other one not)?
- (5) Line 367: I am confused as the authors suddenly state that 'exchange of L704 against F or Y did not abolish CLX export'? But the L704F mutant is the important mutation of this manuscript?
- (6) Again for this section, but also in general – what about the expression levels of these mutants. Are they comparable? In summary, I am very much impressed by the amount of data presented here. All the results support the conclusion, but despite this amount of work I still don't grasp the molecular why. Is it simply a change in interactions? If so, I understand than the dwelling time of the substrate increases in the cavity of the transporter, but that transport is completely (!) abolished is hard to grasp. I therefore have severe doubts that this is the only reason for the observed phenotype. I can only strongly encourage the author to create a mutant that reverts the phenotype. Only this will provide in the end the molecular explanation for this exciting finding.

Reviewer #3

(Remarks to the Author)

ABCG36 / PEN3 plays an important role in plant growth and defense against pathogens by exporting auxin-precursor, Indole3-butyric acid (IBA), and camalexin respectively. The mutant allele *abcg36-5* isolated by (Lu et al., 2015; <https://doi.org/10.1104/pp.15.00182>) was identified to carry an L704F mutation. The IBA response assays and pathogen response/secondary metabolite secretion suggested that L704F could uncouple the transport of IBA and other defense-related compounds (Lu et al., 2015). ABCG36 has been previously shown to transport both IBA and camalexin. It is also known that phosphorylation by QSK1 affects IBA transport (Aryal et al., 2023).

In this paper, Xia J et al., have shown that ABCG36L704F causative mutation in *abcg36-5* affects camalexin transport and not IBA transport supporting the effects of exogenous IBA effect observed by Lu et al., 2015. Through docking simulations, the authors have shown that this residue is essential for substrate-specific binding, and sequence variation using onekp search.

In general, the authors have done extensive transport assays to validate the camalexin transport and have provided clear evidence of to already known function of the L704F aminoacid. Unfortunately, these assays are confirmatory and it would be interesting to show how this residue plays a role during growth vs infection or how from an evolutionary aspect this mechanism allows specificity of different phytoalexin derivatives in defense.

Major concerns:

The plants use ABCG36 to switch between growth and defense. The authors need to clearly describe the significance of why this transport data supporting Lu et al., 2015 is of significance. It is important to determine how the plants use this specificity to switch between growth and defense or is it mainly through QSK1-mediated phosphorylation that allows this switch?

In multiple figures, like in Figure 1d and 1e, the authors mentioned that "Significant differences ($p < 0.05$) of means \pm SE ($n \geq 7$ independent protoplast preparations) were determined using Brown-Forsythe and Welch ANOVA and are indicated by different lowercase letters". But they show what looks like a boxplot which primarily indicates median and quartiles. Did the authors check if the distribution is normal and not skewed and what test was used for that? Also, it is not clear what post-hoc multiple comparison analysis was chosen to determine the letters of significance. For some of the data, the sample size looks unequal, and the appropriate test is needed to validate significance.

Please provide detailed notes in the methods section and the reasons for using them to determine significance.

The authors performed multiple sequence alignments using onekp which is used for checking the conservation in its role in other species for evolutionary analysis. It is not clear what is the main objective of this data in the manuscript. There are different aspects that they could explore to improve the significance of L704 residue like, Different plant families produce various phytoalexins and some are derived from tryptophan which is also a substrate for producing auxin(IAA) during plant growth. The authors could show evidence of camalexin or other close derivatives present in different plant species and how this residue plays a role in specificity towards diverse derivatives in other plant species. Variation of L704 with amino acid shows diverse indolic compounds suggest other closely related transporter orthologs to transport Trp-derived indolic defense compounds or IAA-intermediates. Is the L704 variant "quality control" for switching substrates in other plant families? This information would be of great significance in plant defense vs growth.

Version 1:

Reviewer comments:

Reviewer #1

(Remarks to the Author)

I appreciate the authors' efforts to address my previous comments. With deeper molecular dynamics simulations and mechanistic analysis, the authors have addressed most of my previous concerns, with the exception of the binding assays which might be a impossible task for the authors. The findings of the quality control to the substrate specificity by the key residue identified in ABCG36 by the authors provide interesting perspectives for future studies, not only to the plant field, but general readers of ABC transporters. Therefore, I support the publication of this paper with the revisions, and have no further comments.

Reviewer #2

(Remarks to the Author)

The authors have addressed my points very nicely and I am more than happy with the changes and additions of the revised manuscript. I also understand their answer to point 4 as the timeline of such experiments in plants simply make it impossible for a revision.

In summary I recommend acceptance of the revised version.

Reviewer #3

(Remarks to the Author)

Thanks to the authors for addressing all the comments. The new data and modified text make this work more interesting. The only final suggestion I have is that the evolutionary analysis provides important information about residue conservation in Brassicaceae and will help in designing experiments for the future which makes it ideal to be included in discussion rather than main results. It also depends on the technical ability to detect the both defensive compound and IBA in different species with such sequence conservation in the future.

Reviewer #1 (Remarks to the Author):

The manuscript by Xia et al. presents a deep analysis of the substrate selectivity of the Arabidopsis ABC transporter, ABCG36, as governed by a key residue located along the transport path. Mutation of this residue from leucine to phenylalanine, L704F, reduces its camalexin (CLX) transport activity, while remains its IBA transport activity, which is supported by the extensive plant cell-based transport assay results carried out in the mutant background or the overexpression background. In silico simulations were performed to further support the conclusions. A broader mutagenesis of the key residue further found that the mutant L704Y expands the transport substrates, and allows transport of non-native ligands, including IAA and indole, further supporting that this residue plays essential roles in the substrate specificity of ABCG36. Overall, the authors dissected the effect of the previously identified mutation in the substrate selection of this plant ABC transporter, and helped in understanding how such transporters transport multiple molecules with different chemical structures. Based on presented data, I believe the phenomenon observed and the conclusion reached by the authors are solid. I have several points and suggestions for the authors to consider to improve the quality of the manuscript, particularly regarding the MD simulation and molecular mechanism interpretation part.

We would like to thank R1 for his constructive input and overall positive evaluation!

Major points:

R1.1. When carrying out the transport assays, the protein amount of ABCG36 in the PM should be carefully quantified. It would be difficult to judge the quantity of the WT protein and mutants simply from the confocal imaging results (Fig. 1g, etc.). Using another quantitative method would be much helpful to exclude the possibility that the observed transport differences were due to different protein levels.

We fully agree that this was a shortcoming that needed correction. We have addressed this point by a careful quantification of plasma membrane (PM) signals of confocal images (tobacco work: > 30 individual cells, Arabidopsis work: > 200 individual cells) by using the Fiji plug-in in Image-J, which is a well-accepted tool for confocal image quantification. Quantifications have been added for all images, which concerns Figs. 1g-h, 2e-f and 6e-f. Moreover, we have provided for Arabidopsis material Western blots for Wt and mutant version of ABCG36 using commercial anti-PDR8/ABCG6 and the PM marker, anti-PIP2;1 (see Suppl. Fig. 2j).

Quantifications reveal that individual point mutations did not alter significantly ABCG36 expression levels at the PM, which validates that shown differences in transport are indeed due to the impact of these mutations on transport activity (and not protein expression or stability).

R1.2. The substrate export was calculated by measuring the relative radioactivity in the supernatant. How about directly measure the radioactivity in the cells, which would more faithfully reflect the export process? At least the authors should show this result for the representative control, WT, and L704F control. Since the authors did not describe the assay system in the method, it would be difficult to judge the change in

the supernatant would be significant to reflect the changes. Also the meaning of the Y-axis labelling “% of initial export” is hard to understand, and should be replaced by another phrase.

We are not entirely sure if “protoplast loading” is a better readout for an exporter but agree that such a loading control is meaningful. Therefore, as requested we have performed an uptake/loading experiment for WT ABCG36 and the L704F in comparison to the vector control using tobacco transfection for both IBA and CLX, which is added now as Suppl. Fig. 2c-d. As expected, and in agreement with export experiments these show that IBA loading for L704F is not different to Wt ABCG36, while CLX loading behaves like vector control, indicating that CLX export for L704F is abolished.

*Concerning the Y-label, we prefer to stick to “% of initial export” as it is a very well accepted way of calculating export (or import) from/into entire cells, such as protoplasts, and has been used in numerous publications from my lab and others. As explained in the Methods, relative export from protoplasts is calculated from exported radioactivity taken at t = 0 and 15 min. as follows: (radioactivity in the supernatant at time t = 15 min.) - (radioactivity in the supernatant at time t = 0)) * (100%) / (radioactivity in the supernatant at t = 0 min.). This way of calculation is necessary due to the high background transport that is caused by diffusion of weak organic acids, such as IBA or IAA. The correctness of this calculation has been verified in the original publication establishing this method for IAA (Geisler et al., 2005) by using different means, like protoplast volume, chlorophyll or protein content. In summary, we would like to stick to the Y-label as it is scientifically necessary and correct and well-established in the auxin and plant hormone transport community and above as is also reflected by the following key publications in well-respected journals:*

(Aryal et al., 2023; Chen et al., 2023; Elejalde-Palmett et al., 2021; Geisler et al., 2005; Henrichs et al., 2012; Jarzyniak et al., 2021; Zurcher et al., 2016)

R1.3. In my opinion, the sequence conservation analysis presented in Fig. 5 have limited meaning to understanding the substrate selectivity of ABCG transporters, particularly without the substrate information of the compared transporters. I suggest moving this part into the discussion and SI.

Based on the comments from reviewer 2 and 3 (see comments to R2.1 and R3.3 below), it seems indeed that we failed in explaining why we conducted this part of our work and what the conclusions are. In order to address this point, we have now added a better objective to the Results section (lines 355-365). We assumed that the fact that members of other plant families, like the Fabaceae and Solanaceae, known to employ chemically totally diverse phytoalexins than the Brassicaceae (such as tryptophane derivatives (Brassicaceae), terpenoids (Solanaceae) or isoflavonoids (Fabaceae), respectively) but use an identical set of growth hormones (here: IBA) should be reflected by the variation of key residues of the extracellular gate. Our results indeed indicate that this is the case, and interestingly this evolutionary variation is different for corresponding residues F703/F1347 and L704/F1375 (see Fig. 5, Suppl. Fig. 7).

This special situation of chemically very close substrates in the Brassicaceae family requires in our eyes a special level of quality control provided by the extracellular gate, which as a result has evolutionary separated from that of other families (most obvious for L704) that more or less show a similar degree of amino acid variation. This information has now been added to the Results section (lines 357ff).

According to the request of reviewer 2 (see R2.1) we added additional information on the conservation of a “Gly-hydrophobic-hydrophobic” in plant ABCGs. As can be deduced from the alignment in Suppl. Fig. 7A and the evolutionary analyses in Fig. 5, nearly all plant ABCGs follow this Gly-hydrophobic-hydrophobic pattern, while the first aa is in most cases a Phe. As explained in the text, for the second position we do find substantial variation that itself varies with the plant families. But we do find also a few exceptions, such as Arabidopsis ABCG3 (G-HF) or ABCG33 (G-FT). The overall conservation of the “Gly-hydrophobic-hydrophobic motif” is now added to the text (see lines 372ff).

In light of the serious interest of both reviewers 2 and 3 in the evolutionary analyses and our improved explanations and conclusions, we suggest to keep Fig. 5 as a main figure.

R1.4. I appreciate the authors efforts in characterizing the transport and ATPase activity of the ABCG36 constructs. I wonder if the authors could manage to measure the binding affinity between ABCG36 and the different substrates. It would help to understand how the mutation poses effects to the transport process and which step it might affect.

We had not undertaken binding assays because we were initially not assuming that mutations in the extracellular gate had an effect on substrate binding thought to take place in the substrate binding domain. However, in light of the here reported allosteric cooperativity between extracellular gate and substrate binding domain, we now agree that these binding assays are meaningful, and we thus have conducted them using isolated microsomes prepared from Wt mutant abcg36 Arabidopsis lines. As you can see from the results (new Figs. 4e-f), they clearly support IBA and CLX binding to ABCG36 (significantly reduced binding in the abcg36-4 loss-of-function allele compared to Wt). Unfortunately, despite reductions for the abcg36-6 with both substrates and for the abcg36-5 allele for CLX, all differences for the point mutations were non-significant. Our interpretation (see lines 305ff) is that mutations in the extracellular gate seem to have a long-range effect on transport but that this does only mildly affect drug binding. However, we cannot exclude that the observed reductions in substrate binding are solely insignificant because they are masked by other membrane proteins, including described ABCG-type IBA exporters shown to transport IBA and CLX (see Aryal et al. (2023), for details).

In order to eventually repeat this assay on isolated ABCG36 protein, we contacted Lutz Schmitt from the University of Düsseldorf who had tried to purify ABCG36 (Grafe et al., 2019) for advice. However, we learnt that it is essentially impossible to purify ABCG36 due to its instability (L. Schmitt, pers. information).

R1.5. The MD simulation for the transport process was carried by characterizing the pulling force of the ligand along the transport path. As for ABC transporters, the conformational changes would be the more direct process to exposing the substrate to different side of the PM. It would not be a good system to characterize the substrate selectivity using the pulling force method.

Also the presented results for IBA and CLX in Fig.6f-j are not clear enough in my opinion to fully reveal the difference. It would be more interesting to have ABCG36 models under different transport conformations and compare the substrate binding

profiles for different mutants, and whether L704 directly involved in the substrate binding under certain states.

The mechanism by which ABCG transporters perform substrate translocation remains enigmatic. Notably, there is no experimentally determined structure of a transport-competent conformation. For example, an outward-facing conformation has not been identified and Kaspar Locher from ETH Zurich has proposed a peristaltic mechanism for substrate translocation (Manolaridis et al., 2018). Moreover, a structure with inside-closed transmembrane helices has been observed in the absence of ATP (Orlando and Liao, 2020). These gaps in structural knowledge, and the proposed peristaltic mechanism in our view, justify the use of pulling simulations to probe the translocation pathway.

We fully acknowledge that the force plots (e.g., in Fig. 6) were unclear and difficult to interpret. To address this, we have replaced these panels with structures showing the results of pulling simulations using color encoding to visualize the findings (new Figures 4, 6 and Supplementary Figure 6). These analyses also indicated the direct role of position 704 in substrate interactions.

Unfortunately, we cannot perform simulations with various transport conformations because the key conformations, particularly the outward-facing conformation (if it exists), remain unknown. However, if the peristaltic mechanism is indeed the mode of action, our pulling simulations likely approximate this mechanism, as the protein (including the central helices) undergoes dynamic motions during molecular dynamics simulations.

We have addressed these points in the main text.

Minor points:

1. The x-axis labelling should be rearranged and show more clearly since they are almost overlapped, like in Fig.1h, Fig.3c-d, etc.

Thanks for pointing this out, this has now been corrected.

2. I noticed that the sample size presented in the transport assay result varied for different groups. It would be better to keep them in consistency and further improve the data quality.

This was indeed true for Wt/Vector control samples for that we have usually far more replicates as they are our internal controls. We have conducted and added more transport experiments (which partially also explains the delay for the resubmission), which has increased the data quality (see Figs. 1, 2 and 6) but not the main conclusions.

3. The method part should provide more details of how the experiments are carried out, especially for the transport assay.

The methods, especially for the transport assays, have now been better documented.

Reviewer #2 (Remarks to the Author): PROBABLY LUTZ

The main substrates of the ABC transporter ABCG36 (PEN3 or PDR8) from *A. thaliana* are IBA (indole-3-butyric acid) and CLX (camalexin), respectively. IBA is a precursor of auxin, while CLX is an important antimicrobial compounds. As a member of the ABCG family, ABCG36 likely adopts the fold of human ABGG2. As this, it likely contains a hydrophobic extracellular gate that controls substrate export. A mutant in ABCG36 (L704F) was already described that uncouples the transporter function. To explain this phenotype, it was also suggested that CLX export was impaired while IBA export was similar to wildtype levels. In the submitted manuscript, the authors demonstrate that this is indeed the case. They elaborate this conclusion by different approaches including plant growth experiments, transport assays using protoplast, structural comparison of the ABCG2 structure and an AlphaFold derived model of ABCG36, and various simulation techniques. This is truly an important issue addressed by the authors and the data presented support the conclusion that IBA and CLX transport are uncoupled (IBA at wildtype levels and no CLX transport). I even have to say that I am impressed by the wealth of different methods applied. All these results come to the same conclusion. But my major criticism is the fact that a molecular explanation for such a phenotype is missing. Meaning that presented results remain descriptive in nature.

Below I list my points of concern in the order of appearance, not in the order of importance.

We would like to thank R1 for his constructive input and overall positive evaluation. We have taken his major point – lack of a “molecular explanation” – up seriously and will address our point of view in detail below.

R2.1 It is important to note that the extracellular gate contains the sequence Gly-hydrophobic-hydrophobic in both halves of human ABCG2 or yeast Pdr5. Is a Gly residue also present in ABCG36?

*We had mentioned the conserved glycine in our original submission (lines 368ff) but indeed not commented on its conservation. As can be deduced from the alignment in Suppl. Fig. 7A and the evolutionary analyses in Fig. 5, nearly all plant ABCGs follow this Gly-hydrophobic-hydrophobic pattern, while the first aa is in most cases a Phe. As explained in the text, for the second position we do find substantial variation that itself varies with the plant families. But we do find also a few exceptions, such as *Arabidopsis* ABCG3 (G-HF) or ABCG33 (G-FT). The overall conservation of the “Gly-hydrophobic-hydrophobic motif” is now added to the text (see lines 372ff).*

R2.2 Line 182: I understand why ATPase measurements were performed at pH 9. But is this physiologically meaningful? **Why not bench marking the activity against a sample that contains an inactive ABCG36 version?**

As R2 apparently knows, we measured ATPase at pH 9 in order minimize the strong background activities of ATPase activity of H⁺-ATPases. Obviously, this is not a physiological/cytoplasmic pH; still, we believe that measurements at pH 9 are simply for technical reasons superior.

In any case, we have taken the advice of R2 up and measured ATPase activity of Walker A (K210L) mutations in ABCG36 that are very similar to vector control values (see Fig. 1h).

R2.3 Line 307: Here a first glimpse of a molecular explain for the abrogation of CLX transport is provided – **a stronger interaction of CLX with the transporter**. If this is indeed true, what happens in mutants that reduce these stronger interactions to the IBA situation?

Your comment inspired us to expand our analyses with additional combinations of small molecules and ABCG36 constructs, as well as to implement novel analyses aiming at dissecting the determinants of substrate recognition with greater precision. We do hope that these efforts, along with the accompanying discussion in the text, demonstrate significant progress in understanding and interpreting the factors underlying substrate recognition in ABCGs.

R2.4 Section sequence conservation. If the observed differences in conservation and in the different families has a physiological implication, is it possible to **generate a transporter with a similar phenotype as ABCG36 (one substrate transported, the other one not)**?

After clarification via the editor, we assume that R2 asks in fact for “a mutation in ABCG36 with no or low IBA and high CLX if this is indeed possible”, meaning a reversal of the here described phenotype for L704F. We understand and share the reviewer’s desire to ask for such mutant, however, we would like point out that our knowledge how ABC(G) transporter substrate specificity is encoded is still not sufficient to address such a task directly, even with using the phylogenetic analyses we offer here. In this respect it is important to mention again that the pen3-5/L704F mutation came out of a screen for such an altered substrate-specificity and was as such designed in a “non-targeted manner”. Obviously, such a screen could be conducted by using a similar screening strategy (like root length on toxic concentration of camalexin and IBA, respectively), however, such a complete screen takes with Arabidopsis more than 1 year and is therefore out of the scope of this work. Finally, we would like to mention that in a way our extended mutagenesis in Fig. 6 was partially resulting in such an inverse behavior: L704C transports beside CLX and IBA also IAA but not indole, while L704Y transports all four tested substrates.

R2.5 Line 367: I am confused as the authors suddenly state that ‘exchange of L704 against F or Y did not abolish CLX export’? But the L704F mutant is the important mutation of this manuscript?

We actually meant L704Y and L704W mutations (not L704F), which is obviously not transporting CLX. We are sorry for this flaw that has now been corrected.

R2.6 Again for this section, but also in general – what about the expression levels of these mutants. Are they comparable?

As explained already in point R1.1, we have addressed this point by a careful quantification of plasma membrane (PM) signal of many (tobacco work: > 30 individual cells, Arabidopsis work: > 200 individual cells) confocal images by using the Fiji plugin in Image-J, which is a well-accepted tool for confocal image quantification. Quantifications have been added for all images, which concern Figs. 1g-h, 2e-f and 6e-f. Moreover, we have provided for Arabidopsis material Western blots for Wt and mutant version of ABCG36 using commercial anti-PDR8/ABCG6 and the PM marker, anti-PIP2;1 (see Suppl. Fig. 2j).

Quantifications reveal that individual point mutations did not alter significantly ABCG36 expression levels at the PM, which validates that shown differences in transport are indeed due to the impact of these mutations on transport activity (and not protein expression or stability).

In summary, I am very much impressed by the amount of data presented here. All the results support the conclusion, but despite this amount of work I still don't grasp the molecular why. Is it simply a change in interactions? If so, I understand than the dwelling time of the substrate increases in the cavity of the transporter, but that transport is completely (!) abolished is hard to grasp. I therefore have severe doubts that this is the only reason for the observed phenotype. I can only strongly encourage the author to create a mutant that reverts the phenotype. Only this will provide in the end the molecular explanation for this exciting finding.

We fully understand the reviewer's concerns and have to admit that we were ourselves quite impressed that these point mutations in the extracellular gate lead to a total disruption of CLX transport, while IBA transport is entirely preserved. But we would like to remind the reviewer that this finding on the transport level is absolutely in agreement with the published and here shown growth phenotypes of the pen3-5/L704 mutant (Lu et al., 2015). Moreover, indirect support comes from work on ABCG46 from Medicago (belonging to the Fabaceae family), where F562L (corresponding to F431 in HsABCG2) leads to selective loss of transport of the phenylpropanoid, 4-coumarate, while liquiritigenin is still a functional substrate (Pakuła et al., 2023).

As mentioned already under R2.4, we do also understand the reviewer's desire to have a mutant in hand that reverts the phenotype. Indeed, such a screen could be conducted by using a similar screening strategy, however, such a complete screening that takes with Arabidopsis more than 1 year is in our eyes out of the scope of this work. Instead, we would like to emphasize that in a way our extended mutagenesis in Fig. 6 was partially resulting in such an inverse behavior: L704C transports beside CLX and IBA also IAA but not indole, while L704Y transports all four tested substrates.

As detailed in our responses to R1.4 and R2.3, we have extended our MD simulations and performed more comprehensive analyses. The results from simulations of additional L704 mutations not only reinforce our conclusions but also provide a potential molecular explanation for the observed changes in transport. Specifically, they highlight which residue interactions with the small molecule may facilitate or trigger transport; in addition to direct interactions involving residue 704, dynamic

changes seem to be communicated allosterically to transport pathway residues distant from the gate region

Our interpretation is that mutations in the extracellular gate have a long-range effect on transport but that this does only mildly affect drug binding thought to be provided by the central drug binding site. However, we cannot exclude that the observed reductions in CLX binding for L704F ABCG36 (see Fig. 4f) are solely insignificant because they are masked by other membrane proteins, including described ABCG-type IBA exporters shown to transport IBA and CLX (see Aryal et al. (2023) for details). This is now added to the Results section (lines 305ff).

In summary, our results support an allosteric cooperativity between the extracellular gate and residues in the substrate binding pocket, which can explain the ~~explains~~ described selective transport defects. In our view, this represents a novel and exciting discovery. However, we acknowledge that we were unable to fully elucidate the molecular details of how these point mutations control substrate specificity - an enduring challenge that the ABC transporter community has been struggling with for over 30 years.

Reviewer #3 (Remarks to the Author):

ABCG36 / PEN3 plays an important role in plant growth and defense against pathogens by exporting auxin-precursor, Indole3-butyric acid (IBA), and camalexin respectively. The mutant allele *abcg36-5* isolated by (Lu et al., 2015; <https://doi.org/10.1104/pp.15.00182>) was identified to carry an L704F mutation. The IBA response assays and pathogen response/secondary metabolite secretion suggested that L704F could uncouple the transport of IBA and other defense-related compounds (Lu et al., 2015). ABCG36 has been previously shown to transport both IBA and camalexin. It is also known that phosphorylation by QSK1 affects IBA transport (Aryal et al., 2023).

In this paper, Xia J et al., have shown that ABCG36L704F causative mutation in *abcg36-5* affects camalexin transport and not IBA transport supporting the effects of exogenous IBA effect observed by Lu et al., 2015. Through docking simulations, the authors have shown that this residue is essential for substrate-specific binding, and sequence variation using onekp search.

In general, the authors have done extensive transport assays to validate the camalexin transport and have provided clear evidence of already known function of the L704F amino acid. Unfortunately, these assays are confirmatory and it would be interesting to show how this residue plays a role during growth vs infection or how from an evolutionary aspect this mechanism allows specificity of different phytoalexin derivatives in defense.

We would like to thank R3 for his constructive input and overall positive evaluation. Before going into details, we would like to mention that the main purpose of this study was - beside the verification of suggestive transport defects described in Lu et al. (2015) - to find a mechanistic explanation why or how these mutations affect unilaterally substrate transport. We have done so by using sophisticated MD analyses, which were even extended in the new version of the manuscript, and partially also by new binding assays. As such we provide for the first time a mechanistic rationale for the transport discrimination of CLX by showing, which is caused by an increase in free

energy and pulling forces for CLX at both the entrance and exit sites of ABCG36^{L704F}, respectively. Therefore, we believe that L704 indeed is key in the regulation of growth-defense balance decisions via ABCG36. The key role of ABCG36 in this process is also indirectly supported by the here newly uncovered putative signaling function of ABCG36 during infection. In summary, we believe that our study indeed contributes to our understanding “how this residue plays a role during growth vs infection”. We would like to mention that the evolution of these key residues of the extracellular gate was indeed only a minor – but as explained below - an important aspect of this study that might form the basis for future studies.

Major concerns:

R3.1 The plants use ABCG36 to switch between growth and defense. The authors need to clearly describe the significance of why this transport data supporting Lu et al., 2015 is of significance. It is important to determine how the plants use this specificity to switch between growth and defense or is it mainly through QSK1-mediated phosphorylation that allows this switch?

These are indeed two very crucial points that we indeed did not explain well and that require further explanation. We have now better described the role of ABCG36 during growth-defense tradeoff decisions in the Introduction and Discussion section (see lines 86ff and 595ff, respectively). Please note that we do not use the word “switch” as ABCG36 phosphorylation represses unilaterally IBA export allowing for camalexin transport.

Obviously, L704 mutation is a somehow “artificial situation” but we believe that based on our data presented in Aryal et al. (2023) and here, it is indeed reasonable to assume that ABCG36 phosphorylation by QSK1 can cause long-range movements of L704 and F1375 (and neighbor residues) in the extracellular gate allowing for altered substrate selectivity. Obviously, such phosphorylation events can have also an effect on substrate binding in the central cavity. Both options might be even interconnected based on our found allosteric cooperativity between extracellular gate and substrate binding domains; a dissection of the underlying mechanisms is currently underway. In any case, a puzzling question is how phosphorylation of a single serine on obvious flexible and disordered regions, such as the N-terminus and the linker region of ABCG36, can have such a strong effect on ABCG36 substrate specificity. One explanation might be that these regions function in analogy to the regulatory linker or R-domains connecting the two halves of full-size transporters of ABCBs or CYSTIC FIBROSIS TRANSMEMBRANE CONDUCTANCE REGULATOR (CFTR)/ABCC7, respectively. Here, phosphorylation is thought to add negative charges that by means of conformational changes alter ABC functional capacities (Aryal et al., 2015; Gadsby et al., 2006; Geisler and Hegedus, 2020). This model was very recently supported by cryo-EM imaging of the ABCC, YEAST CADMIUM FACTOR (YCF1), showing that its R-domain associates upon phosphorylation with NBD1 adjusting directly its ATPase activity (Khandelwal et al., 2022). For yeast PDRs/ABCGs, it was shown that NBDs influence transport activity but also substrate specificity (Kolaczowski et al., 2013). In our eyes, such a long-range effect is eventually not caused directly by phosphorylation of these flexible loops but indirectly by allowing interacting, regulatory proteins to bind to these surfaces. In this model, both phosphorylated clusters form surfaces for protein-protein interactions with regulatory proteins or other transporters

as described for the R-domain of CFTR/ABCC7 (Bozoky et al., 2013). This overall explanation is now added in brief to the Discussion section (lines 603ff).

R3.2 In multiple figures, like in Figure 1d and 1e, the authors mentioned that “Significant differences ($p < 0.05$) of means \pm SE ($n \geq 7$ independent protoplast preparations) were determined using Brown-Forsythe and Welch ANOVA and are indicated by different lowercase letters”. But they show what looks like a boxplot which primarily indicates median and quartiles. Did the authors check if the distribution is normal and not skewed and what test was used for that? Also, it is not clear what post-hoc multiple comparison analysis was chosen to determine the letters of significance. For some of the data, the sample size looks unequal, and the appropriate test is needed to validate significance. Please provide detailed notes in the methods section and the reasons for using them to determine significance.

It is correct that we used so-called boxplots throughout the figures that in our eyes is the most precise and best-readable presentation of our data (and also requested by the journal guidelines). As such they indeed show median and 25th and 75th percentiles represented by the box itself and the middle line, respectively. However, statistics were as previously correctly annotated calculated on means that are now indicated by a “+”. Throughout the ms. either “Ordinary One-way ANOVA” or “Two-way ANOVA” and “Tukey’s multiple comparison test” was used to determine significances. Normal distribution of all data prior to statistical analyses was already analyzed in the previous version of the ms. by using the Prism-embedded “D’Agostino-Pearson omnibus normality test”. This information is now indicated in the methods and/or legends, respectively.

Sample size has now been adjusted (by adding substantial transport repetitions), which was also requested by R1.

R3.3 The authors performed multiple sequence alignments using onekp which is used for checking the conservation in its role in other species for evolutionary analysis. It is not clear what is the main objective of this data in the manuscript. There are different aspects that they could explore to improve the significance of L704 residue like, Different plant families produce various phytoalexins and some are derived from tryptophan which is also a substrate for producing auxin (IAA) during plant growth. The authors could show evidence of camalexin or other close derivatives present in different plant species and how this residue plays a role in specificity towards diverse derivatives in other plant species.

Variation of L704 with amino acid shows diverse indolic compounds suggest other closely related transporter orthologs to transport Trp-derived indolic defense compounds or IAA-intermediates. Is the L704 variant “quality control” for switching substrates in other plant families? This information would be of great significance in plant defense vs growth.

It seems indeed that we failed in explaining why we conducted this part of our work and what the conclusions are. We have now added a better objective to the Results section (lines 357ff). Members of other plant families, like the Fabaceae (isoflavonoids) and Solanaceae (terpenoids), are known to employ chemically diverse phytoalexins

than the Brassicaceae (indolic compounds that are tryptophane derivatives) but use an identical set of growth hormones (here: IBA), which is also an indole. Based on such a chemical overlap between transported defense and growth substances we proposed that the extracellular gate is an important but supportive determinant of selectivity and may vary according to the metabolite/defensive compound profile. Our results indeed indicate that this is the case, and interestingly this evolutionary variation is different for corresponding residues F703/F1347 and L704/F1375 (see Fig. 5 and Suppl. Fig. 7). In summary, we do not believe that the key residues of the extracellular gate are the primary determinants of substrate specificity in ABC(G)s (these are most-likely the substrate binding domains in the central cavity) but that in the Brassicaceae family ABCGs are confronted with chemically very similar but physiologically very distinct substrates, because both IBA as a growth hormone and camalexin (and other indoles) as defense compounds are both indolic compounds, while in other families they are chemically distinct. Of note, Brassicaceae have evolved the capacity to produce thioglucosides (also known as glucosinolates), which are also indole-type phytoalexins. Obviously, this requires in our eyes a special level of quality control provided by the extracellular gate families (most obvious for L704), which as a result has evolutionary separated from that of other that more or less show a similar degree of amino acid variation. This information as well as the metabolic connection between Trp-derived indolic defense compounds or IAA-intermediates has now been added to the Results and Discussion sections (lines 357ff and 547ff).

References

- Aryal B, Laurent C, Geisler M.** 2015. Learning from each other: ABC transporter regulation by protein phosphorylation in plant and mammalian systems. *Biochem Soc Trans* **43**, 966-974.
- Aryal B, Xia J, Hu Z, Stumpe M, Tsering T, Liu J, Huynh J, Fukao Y, Glockner N, Huang HY, Sancho-Andres G, Pakula K, Ziegler J, Gorzolka K, Zwiewka M, Nodzynski T, Harter K, Sanchez-Rodriguez C, Jasinski M, Rosahl S, Geisler MM.** 2023. An LRR receptor kinase controls ABC transporter substrate preferences during plant growth-defense decisions. *Curr Biol* **33**, 2008-2023 e2008.
- Bozoky Z, Krzeminski M, Chong PA, Forman-Kay JD.** 2013. Structural changes of CFTR R region upon phosphorylation: a plastic platform for intramolecular and intermolecular interactions. *FEBS J* **280**, 4407-4416.
- Chen J, Hu Y, Hao P, Tsering T, Xia J, Zhang Y, Roth O, Njo MF, Sterck L, Hu Y, Zhao Y, Geelen D, Geisler M, Shani E, Beeckman T, Vanneste S.** 2023. ABCB-mediated shootward auxin transport feeds into the root clock. *EMBO Rep* **24**, e56271.
- Elejalde-Palmett C, Martinez San Segundo I, Garroum I, Charrier L, De Bellis D, Mucciolo A, Guerault A, Liu J, Zeisler-Diehl V, Aharoni A, Schreiber L, Bakan B, Clausen MH, Geisler M, Nawrath C.** 2021. ABCG transporters export cutin precursors for the formation of the plant cuticle. *Curr Biol* **31**, 2111-2123 e2119.

Gadsby DC, Vergani P, Csanady L. 2006. The ABC protein turned chloride channel whose failure causes cystic fibrosis. *Nature* **440**, 477-483.

Geisler M, Blakeslee JJ, Bouchard R, Lee OR, Vincenzetti V, Bandyopadhyay A, Titapiwatanakun B, Peer WA, Bailly A, Richards EL, Ejendal KF, Smith AP, Baroux C, Grossniklaus U, Muller A, Hrycyna CA, Dudler R, Murphy AS, Martinoia E. 2005. Cellular efflux of auxin catalyzed by the Arabidopsis MDR/PGP transporter AtPGP1. *Plant J* **44**, 179-194.

Geisler M, Hegedus T. 2020. A twist in the ABC: regulation of ABC transporter trafficking and transport by FK506-binding proteins. *FEBS Lett* **594**, 3986-4000.

Grafe K, Shanmugarajah K, Zobel T, Weidtkamp-Peters S, Kleinschrodt D, Smits SHJ, Schmitt L. 2019. Cloning and expression of selected ABC transporters from the Arabidopsis thaliana ABCG family in Pichia pastoris. *PLoS One* **14**, e0211156.

Henrichs S, Wang B, Fukao Y, Zhu J, Charrier L, Bailly A, Oehring SC, Linnert M, Weiwad M, Endler A, Nanni P, Pollmann S, Mancuso S, Schulz A, Geisler M. 2012. Regulation of ABCB1/PGP1-catalysed auxin transport by linker phosphorylation. *EMBO J* **31**, 2965-2980.

Jarzyniak K, Banasiak J, Jamruszka T, Pawela A, Di Donato M, Novak O, Geisler M, Jasinski M. 2021. Early stages of legume-rhizobia symbiosis are controlled by ABCG-mediated transport of active cytokinins. *Nat Plants* **7**, 428-436.

Khandelwal NK, Millan CR, Zangari SI, Avila S, Williams D, Thaker TM, Tomasiak TM. 2022. The structural basis for regulation of the glutathione transporter Ycf1 by regulatory domain phosphorylation. *Nat Commun* **13**, 1278.

Kolaczkowski M, Sroda-Pomianek K, Kolaczowska A, Michalak K. 2013. A conserved interdomain communication pathway of pseudosymmetrically distributed residues affects substrate specificity of the fungal multidrug transporter Cdr1p. *Biochim Biophys Acta* **1828**, 479-490.

Lu X, Dittgen J, Pislewska-Bednarek M, Molina A, Schneider B, Svatos A, Doubisky J, Schneeberger K, Weigel D, Bednarek P, Schulze-Lefert P. 2015. Mutant Allele-Specific Uncoupling of PENETRATION3 Functions Reveals Engagement of the ATP-Binding Cassette Transporter in Distinct Tryptophan Metabolic Pathways. *Plant Physiol* **168**, 814-827.

Manolaridis I, Jackson SM, Taylor NMI, Kowal J, Stahlberg H, Locher KP. 2018. Cryo-EM structures of a human ABCG2 mutant trapped in ATP-bound and substrate-bound states. *Nature* **563**, 426-430.

Orlando BJ, Liao M. 2020. ABCG2 transports anticancer drugs via a closed-to-open switch. *Nat Commun* **11**, 2264.

Zurcher E, Liu J, di Donato M, Geisler M, Muller B. 2016. Plant development regulated by cytokinin sinks. *Science* **353**, 1027-1030.

Reviewer #3 (Remarks to the Author):

Thanks to the authors for addressing all the comments. The new data and modified text make this work more interesting. The only final suggestion I have is that the evolutionary analysis provides important information about residue conservation in Brassicaceae and will help in designing experiments for the future which makes it ideal to be included in discussion rather than main results. It also depends on the technical ability to detect the both defensive compound and IBA in different species with such sequence conservation in the future.

Thanks for the overall positive evaluation of our revision.

In principle we do understand your request but feel that it is better to leave the descriptive part of this figure in the Results as it would artificially blow up the Discussion section.

However, we have accommodated your request as follows:

1. We have exchanged Fig. 5 and 6, leading now to the fact that the evolutionary part is now at the very end of the ms.
2. As a consequence, we have presented now the part that describes this evolutionary aspect (new Fig. 6) at the end of the Results part.
3. And obviously, we end now also in the Discussion section with this part, as originally suggested by you.

In summary, we do hope that you can agree with us that we have addressed your final comment accordingly.